# Genomic rearrangements generate hypervariable mini-chromosomes in host-specific isolates of the blast fungus

Thorsten Langner[1], Adeline Harant[1], Luis B. Gomez-Luciano[2¤a], Ram K. Shrestha[1], Angus Malmgren[1], Sergio M. Latorre[3¤b], Hernán A. Burbano[4], Joe Win[1], Sophien Kamoun[1]*

1 The Sainsbury Laboratory, University of East Anglia, Norwich Research Park, Norwich, United Kingdom, 2 Biodiversity Research Center, Academia Sinica, Taipei, Taiwan, 3 Research group for Ancient Genomics and Evolution, Department of Molecular Biology, Max Planck Institute for Developmental Biology, Tübingen, Germany, 4 Centre for Life's Origins and Evolution, Department of Genetics, Evolution and Environment, University College London, London, United Kingdom

¤a Current address: Universidad Católica del Cibao (UCATECI), La Vega, Dominican Republic
¤b Current address: Centre for Life's Origins and Evolution, Department of Genetics, Evolution and Environment, University College London, London, United Kingdom
* sophien.kamoun@tsl.ac.uk

**Data Availability Statement:** The authors confirm that all data underlying the findings are fully available without restriction. All sequence data

## Abstract

Supernumerary mini-chromosomes–a unique type of genomic structural variation–have been implicated in the emergence of virulence traits in plant pathogenic fungi. However, the mechanisms that facilitate the emergence and maintenance of mini-chromosomes across fungi remain poorly understood. In the blast fungus *Magnaporthe oryzae* (Syn. *Pyricularia oryzae*), mini-chromosomes have been first described in the early 1990s but, until very recently, have been overlooked in genomic studies. Here we investigated structural variation in four isolates of the blast fungus *M. oryzae* from different grass hosts and analyzed the sequences of mini-chromosomes in the rice, foxtail millet and goosegrass isolates. The mini-chromosomes of these isolates turned out to be highly diverse with distinct sequence composition. They are enriched in repetitive elements and have lower gene density than core-chromosomes. We identified several virulence-related genes in the mini-chromosome of the rice isolate, including the virulence-related polyketide synthase *Ace1* and two variants of the effector gene *AVR-Pik*. Macrosynteny analyses around these loci revealed structural rearrangements, including inter-chromosomal translocations between core- and mini-chromosomes. Our findings provide evidence that mini-chromosomes emerge from structural rearrangements and segmental duplication of core-chromosomes and might contribute to adaptive evolution of the blast fungus.

## Author summary

The genomes of plant pathogens often exhibit an architecture that facilitates high rates of dynamic rearrangements and genetic diversification in virulence associated regions.

used in this study was deposited at the European Nucleotide Archive (ENA, https://www.ebi.ac.uk/ena/browser/home) with the study accession PRJEB27137. Nanopore sequencing reads were deposited under accession numbers ERR2612751 (BR32), ERR2612749 (FR13), ERR2612750 (US71), and ERR2612752 (CD156). Nanopore sequence assemblies were deposited under accession numbers GCA_900474545.2 (BR32), GCA_900474655.2 (FR13), GCA_900474175.2 (US71), and GCA_900474475.2 (CD156). Polished whole genome assemblies were deposited under accession numbers GCA_900474545.3 (BR32), GCA_900474655.3 (FR13), GCA_900474175.3 (US71), and GCA_900474475.3 (CD156). Mini-chromosome isolation sequencing reads were deposited under accession numbers ERR3771227-ERR3771238.

**Funding:** This work was supported by the European Research Council (ERC proposal BLASTOFF 743165), the Biotechnology and Biological Sciences Research Council (BBSRC ISP Plant Health), and the Gatsby Charitable Foundation (TSL). The funders had no role in study design, data collection and analysis, decision to publish, or preparation of the manuscript.

**Competing interests:** The authors have declared that no competing interests exist.

These regions, which tend to be gene sparse and repeat rich, are thought to serve as a cradle for adaptive evolution. Supernumerary chromosomes, i.e. chromosomes that are only present in some but not all individuals of a species, are a special type of structural variation that have been observed in plants, animals, and fungi. Here we identified and studied supernumerary mini-chromosomes in the blast fungus *Magnaporthe oryzae*, a pathogen that causes some of the most destructive plant diseases. We found that rice, foxtail millet and goosegrass isolates of this pathogen contain mini-chromosomes with distinct sequence composition. All mini-chromosomes are rich in repetitive genetic elements and have lower gene densities than core-chromosomes. Further, we identified virulence-related genes on the mini-chromosome of the rice isolate. We observed large-scale genomic rearrangements around these loci, indicative of a role of mini-chromosomes in facilitating genome dynamics. Taken together, our results indicate that mini-chromosomes contribute to genome rearrangements and possibly adaptive evolution of the blast fungus.

## Introduction

Genomes of plant pathogens are highly dynamic and typically exhibit an architecture that facilitates rapid adaptation to their hosts. Since the rise of genome sequencing it became evident that plant pathogen genomes are often structured to facilitate high genetic diversification rates at virulence-related loci while maintaining relative stability in house-keeping regions, a phenomenon that shaped the term "two-speed genome" [1]. Since then, the two-speed genome concept has been widely documented in a number of plant pathogenic species [2,3]. Interestingly, various types of genome architecture have been observed in different species. These include effector gene clusters [4,5], lineage-specific genomic regions that are rich in transposable elements [6–10], or enrichment of virulence related genes in specific genomic regions, e.g. unstable telomeric and sub-telomeric regions [11]. Typically, these genomic compartments display higher rates of adaptive mutations compared to the rest of the genome [12]. In addition to signatures of single nucleotide polymorphisms (SNPs) indicative of positive selection and presence/absence polymorphisms, structural variation is common in pathogen populations and ranges from copy number variations of single genes to chromosome-scale rearrangements [13,14]. Extreme cases of genomic rearrangement are large-scale, chromosome length variations and the presence of isolate-specific, supernumerary chromosomes (syn. B-, accessory-, conditionally dispensable, mini-chromosomes). These are usually small, non-essential chromosomes that occur in addition to the regular set of conserved chromosomes within a species and have been described in animals, plants, and fungi [14,15]. Supernumerary chromosomes are present at different frequencies in natural populations of eukaryotic plant pathogens [14,16]. In some fungal species, supernumerary chromosomes have been directly implicated in the emergence of new virulence traits, underpinning the importance of understanding their role in evolution and pathogen adaptation [17–19]. However, the diversity of supernumerary chromosomes across plant pathogens and their contribution to genome plasticity is still poorly known.

 Supernumerary chromosomes share common features that distinguish them from core-chromosomes. Although they are variable in size, they tend to be smaller than core-chromosomes ranging from ~400 kb to 3 Mb in plant pathogenic fungi [8,14]. Yet, despite their small size, supernumerary chromosomes can contribute as much as 15% to the total genome in certain species. Their number is also variable with some plant pathogenic fungi containing up to eight supernumerary chromosomes in addition to the core-chromosome set [9]. Dynamic loss

or gain of supernumerary chromosomes has been observed especially under stress conditions indicating that supernumerary chromosomes are major drivers of genome plasticity [20]. In addition to frequent genomic rearrangements, supernumerary chromosomes often do not follow Mendelian inheritance. They tend to be meiotically unstable, and thus, are frequently lost [21]. However, in some cases meiotic gene drives have been observed increasing their capacity to be inherited and potentially explaining their abundance in pathogen populations [22].

Supernumerary chromosomes are usually gene poor and repeat rich compared to the core-chromosomes, and form one illustration of the two-speed genome concept [12,23]. However, a clear association with adaptive evolution is not always evident as they do not necessarily carry virulence-related genes [9,24]. Nonetheless, in some plant pathogenic fungi, supernumerary chromosomes are directly implicated in virulence [17,18]. The origin of supernumerary chromosomes is still debated but there is evidence for segmental duplications from core-chromosomes (or ancient core-chromosome duplication followed by partial chromosome loss) and horizontal chromosome transfer [19,25–27]. Interestingly, supernumerary chromosomes can be transferred between isolates independently of the core genome and can alter the virulence spectrum of plant pathogens [19,27]. It is thus possible that supernumerary chromosomes facilitate gene flow in natural populations leading to new pathotypes.

Plant pathogens are ubiquitous in the environment and can cause severe damage to both cultivated and wild plant species [28–31]. Plant pathogens are often specialized on specific host species or taxon. At the center of the co-evolutionary dynamics between pathogens and plants are effector proteins, i.e. secreted proteins that manipulate host processes to facilitate infection and colonization [32–35]. In return, host plants have evolved immune receptors that can detect conserved molecular patterns and effector proteins to defend against invading pathogens [36,37]. This generally leads to fast-paced co-evolution between pathogens and their host plants that often follows arms-race dynamics where the frequency of adaptive mutations rises quickly in pathogen populations [16].

The fungus *M. oryzae* (Syn. *Pyricularia oryzae*) causes blast disease, one of the most devastating crop diseases worldwide resulting in yield losses in rice and wheat that make it a threat to global food security [28,29,38,39]. Despite its Linnaean binomial name, *M. oryzae* is a multihost pathogen that can infect more than 50 cultivated and wild grass species. Population genomics of *M. oryzae* revealed that the species is formed by an assemblage of differentiated lineages that are associated with particular host taxa, such as important cereals like rice (*Oryza sativa*), finger millet (*Eleusine coracana*), wheat (*Triticum aestivum*), and foxtail millet (*Setaria italica*), as well as weeds such as Indian goosegrass (*Eleusine indica*) [40,41]. Records of rice blast disease in China date back to the early 17th century and until today it is recognized as one of the most threatening and widely distributed rice diseases [39]. Recent population genetics studies revealed that the rice-infecting lineage of *M. oryzae* consists of both a recombining population and multiple, clonally expanded lineages that have emerged in the last few hundred years [42–44]. In Europe, the rice blast fungus population consists of only one of three major clonal lineages and it is possible that mating type isolation led to local asexual expansions. Another impactful blast disease is wheat blast. In the mid 1980s the disease emerged on wheat plants in Brazil and has since spread across large regions of South America and, more recently, South Asia, demonstrating the ability of *M. oryzae* to rapidly undergo host-range expansions and global pandemics [45–47].

Although signatures of gene flow have been observed within and between lineages [41], *M. oryzae* is thought to predominantly propagate asexually in agricultural settings. Given that genetically differentiated lineages tend to be associated with particular host genera, selection pressure imposed by the host plant is probably the main driver of adaptive evolution [41]. Adaptation to a specific host can be accompanied by gain or loss of effector genes that define the pathogen host range highlighting the importance of structural genomic variation [35,48–

50]. The degree to which genome architecture facilitates structural variation and even gene flow is a fascinating but still poorly understood question [6,11,23,51].

*M. oryzae* mini-chromosomes have first been described in the early 1990s, when studies on karyotype diversity revealed that large chromosomal rearrangements occur frequently within and between clonal lineages [52,53]. More recently, Chuma *et al.* [11] demonstrated that the effector gene *AVR-Pita* underwent multiple translocation events and proposed that genomic location of effector genes to sub-telomeric regions could favor rearrangements within the genome and gene flow within asexual populations. Interestingly, *AVR-Pita* genes occur on different chromosomes, including supernumerary mini-chromosomes. Additionally, Luo *et al.* [54] and Kusaba *et al.* [55] showed that two variants of the AVR-Pik effector gene are present on a 1.6 Mb mini-chromosome in the Japanese rice blast isolate 84R-62B and that parts of this mini-chromosome can translocate to core-chromosomes in crosses. Further, loss of the mini-chromosome resulted in gain of virulence on host plants carrying the rice immune receptor Pik, a phenotype due to the associated loss of the *AVR-Pik* genes [55].

Despite the fact that *M. oryzae* mini-chromosomes have been known for 30 years, genome sequencing projects have somehow overlooked them. This has changed recently. Peng *et al.* [56] reported the first sequences of mini-chromosomes of *M. oryzae*. They analyzed the karyotypes of the three wheat blast isolates T25, B71, and P3. T25 was collected in Brazil in 1988 and B71 and P3 were collected in 2012 in Brazil and Paraguay, respectively. B71 contains a 2 Mb mini-chromosome and P3 contains two mini-chromosomes of 1.5 and 3 Mb, whereas T25 does not contain any mini-chromosomes. The mini-chromosomes of B71 and P3 only share partial sequence similarity, have lower gene and higher repeat content, and display partial similarity to sub-telomeric regions of core-chromosomes. The mini-chromosome of isolate B71 contains the effector genes *Pwl2* and *Bas1* in close proximity whereas they are located on separate core-chromosomes in other *M. oryzae* isolates. This raised the hypothesis that mini-chromosomes are sites of structural rearrangements associated with virulence factors within blast genomes. However, the genetic diversity of mini-chromosomes in other lineages of *M. oryzae*, and their association with genomic rearrangements and effector diversification remains poorly understood.

The objective of this study was to gain an overall picture of genomic structural variation across lineages of *M. oryzae* using the host-specific isolates from rice, foxtail millet, goosegrass and wheat that were previously sequenced using Illumina short reads by Chiapello et al. [48]. Our analyses led us to focus on mini-chromosomes, which we detected in three of the examined isolates of *M. oryzae*. We used long and short read sequencing data in combination with mini-chromosome isolation sequencing (MCIS) to improve the previous genome assemblies [48] to near chromosome quality. We found that the sequence composition of mini-chromosomes is highly variable indicating independent emergence of individual mini-chromosomes or rapid divergence including gene presence/absence polymorphisms during *M. oryzae* lineage evolution. Further, we identified effector and virulence-related genes in the mini-chromosome of the rice-infecting isolate FR13 although this protein class was not generally enriched in the three sequenced mini-chromosomes compared to the core chromosomes. We documented several structural rearrangements around virulence-related loci which raises the possibility that chromosome-scale variation, notably mini-chromosome genesis, plays a role in driving adaptive genome plasticity in the blast fungus.

## Results

### Near chromosome quality genome assemblies of four host-specific isolates of *M. oryzae*

Considering that structural genomic variation has emerged as a common feature of fungal plant pathogens, we re-examined 4 previously sequenced *M. oryzae* genomes of the host

specific isolates FR13 (rice), US71 (foxtail millet), CD156 (goosegrass), and BR32 (wheat). We used long read sequencing to generate highly contiguous assemblies [57] and improved accuracy of the assemblies by applying a polishing pipeline using nanopore and published Illumina raw reads of the same strains [48]. To assess the completeness and quality of our assemblies, we performed a benchmarking universal single-copy orthologs (BUSCO) analysis using the Sordariomycota database (https://busco.ezlab.org/) which confirmed 98–98.2% completeness, similar to the chromosome quality reference genome of strain 70–15 (assembly version 8, ensemble release 45; 98.2%; Table 1) [58]. Overall, the new assemblies have vastly improved contiguity with scaffold numbers reduced from 111–2,051 in previous assemblies to 17–55 in the new assemblies (Table 1). Moreover, we increased the proportion of well-assembled repeat rich regions, reduced the number of ambiguous bases ("Ns") to zero, and improved overall completeness (Table 1). These highly contiguous assemblies can thus serve as new reference genomes for host-specific isolates of *M. oryzae*.

## The rice, foxtail millet and goosegrass isolates of *M. oryzae* contain unique sets of mini-chromosomes

To assess the genomes for karyotype variation, we separated and visualized intact chromosomes of each strain by contour-clamped homogenous electric field (CHEF) gel electrophoresis. Strikingly, we found large-scale, structural variation in the form of chromosome length polymorphisms and supernumerary mini-chromosomes in the isolates FR13, US71, and CD156 (Fig 1). The mini-chromosomes ranged in size between approximately 800 kb and 1.5 Mb. We hypothesized that these mini-chromosomes contribute to structural variation and might be similar to supernumerary, lineage specific chromosomes reported in other plant pathogenic fungi. Supernumerary chromosomes can have important functions in pathogenicity, but their sequence composition and intraspecies variation in *M. oryzae* is still poorly understood.

**Table 1. Comparative summary statistics of genome assemblies.**

| Host | *Oryza sativa* | | *Setaria italica* | | *Eleusine indica* | | *Triticum sp.* | | *Oryza sativa* |
|---|---|---|---|---|---|---|---|---|---|
| Isolate | FR13 GEMO* | FR13 | US71 GEMO* | US71 | CD156 GEMO* | CD156 | BR32 GEMO* | BR32 | 70–15 |
| Technology | Illumina / 454 | ON-MinION | Illumina / 454 | ON-MinION | Illumina / 454 | ON-MinION | Illumina / 454 | ON-MinION | Sanger |
| Coverage | 4x | 174x | 80x | 107x | 50x | 80x | 55x | 131x | N/A |
| # Contigs | 79,619 | 46 | 7,398 | 84 | 26,535 | 44 | 6,044 | 21 | N/A |
| # Scaffolds | 2,051 | 31 | 220 | 55 | 237 | 27 | 111 | 17 | 8 |
| Assembly size (bp) | 43065003 | 46455514 | 41206925 | 45614181 | 42691742 | 43975886 | 41858488 | 41851079 | 41027733 |
| Largest scaffold | 557675 | 7382384 | 3113975 | 6027169 | 2814965 | 8365714 | 4783357 | 11480887 | 8319966 |
| N50 (bp) | 101645 | 5398440 | 813981 | 2812411 | 1066457 | 5531649 | 1760460 | 5096353 | 6606598 |
| N75 (bp) | 37427 | 2617860 | 277145 | 1350202 | 408473 | 3948849 | 837619 | 3936184 | 4490059 |
| L50 | 124 | 4 | 12 | 5 | 13 | 4 | 7 | 3 | 3 |
| L75 | 290 | 7 | 33 | 11 | 31 | 6 | 16 | 5 | 5 |
| BUSCO | 60.90% | 98.10% | 98.10% | 98.10% | 98.00% | 98.00% | 97.70% | 98.00% | 98.20% |
| Repeat content | 1.68% | 13.23% | 3.29% | 13.23% | 2.77% | 6.21% | 3.25% | 6.26% | 11.54% |
| GC-content | 51.39 | 50.99 | 51.09 | 50.65 | 50.96 | 49.87 | 50.81 | 50.06 | 51.61 |
| % N | 22.55 | 0.00 | 5.45 | 0.00 | 6.59 | 0.00 | 4.96 | 0.00 | 0.19 |
| # N's per 100 kbp | 22547.58 | 0 | 5445.7 | 0 | 6589.36 | 0 | 4962.16 | 0 | 189.63 |

* from Chiapello *et al*., 2015 [48]

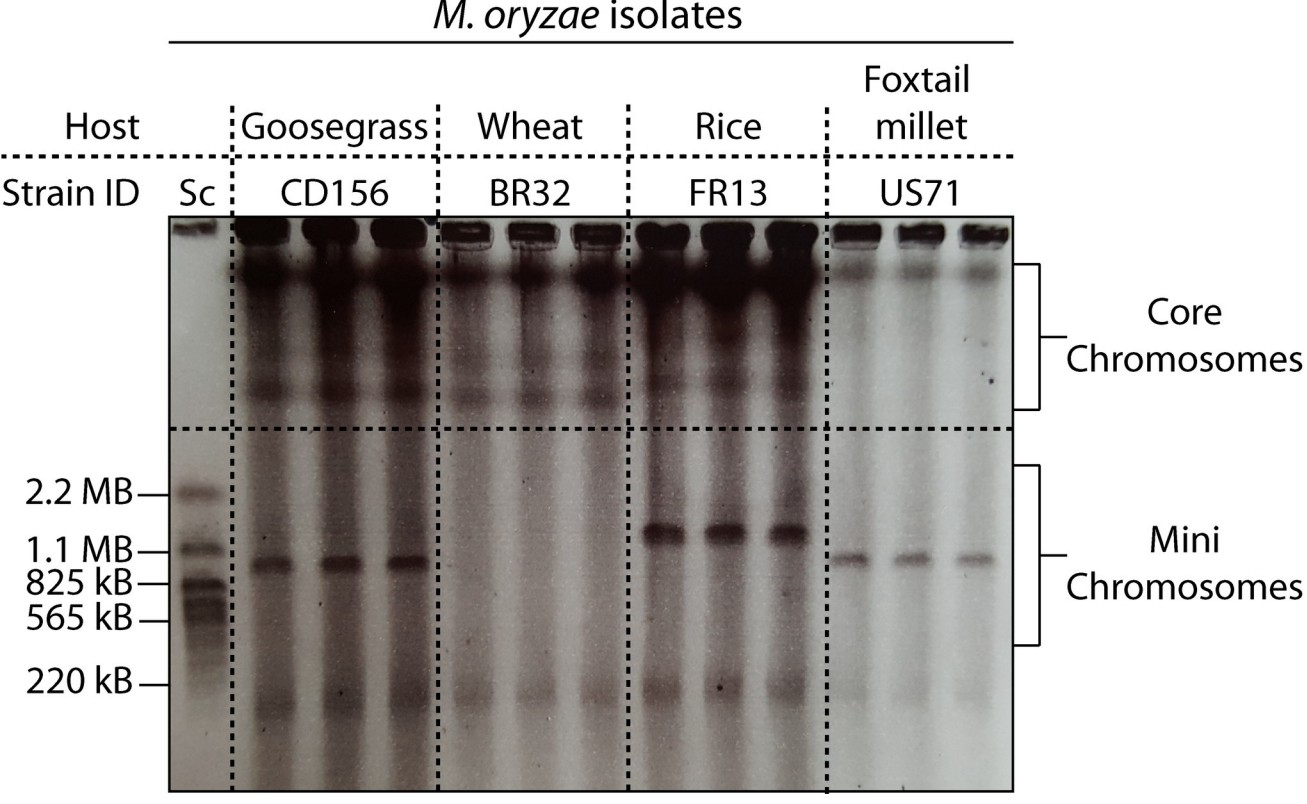

**Fig 1. Host-specific isolates of *M. oryzae* contain mini-chromosomes of various sizes.** Contour-clamped homogenous electric field (CHEF) gel electrophoresis of intact *M. oryzae* chromosomes. Chromosome size variation between *M. oryzae* isolates is present in both, mini-chromosomes and core chromosomes. Strains FR13, US71, and CD156 contain mini-chromosomes ranging in size between approximately 850 kb and 1.5 Mb. Left lane: *Saccharomyces cerevisiae* chromosomes as size marker.

To further analyze the sequence composition of the observed mini-chromosomes, we identified their corresponding scaffolds in our whole genome assemblies by purifying mini-chromosomal DNA from CHEF gels by electro-elution for mini-chromosome isolation sequencing (detailed protocol available on protocol.io [57]; adapted from [56]). We then mapped mini-chromosome derived raw reads against the whole genome assemblies to identify specific scaffolds with high coverage compared to the rest of the genome, indicating their mini-chromosomal origin (Figs 2 and S1–S4). Importantly, all core-chromosome scaffolds (> 2 Mb) showed extremely low coverage confirming the robustness of this approach (S1–S4 Figs).

Repetitive sequences could lead to ambiguous mapping of reads due to the presence of similar repetitive regions in core-chromosomes. To confirm that the observed increase in coverage is linked to mini-chromosome enrichment and not to ambiguous read mapping, we plotted the repeat content per 10 kb sliding window and compared it to the coverage of uniquely mapping reads derived from mini-chromosome sequencing. This analysis showed that repeat-rich regions did not overlap with regions of high coverage, which confirmed that the enrichment was not due to repetitive sequences and ambiguous read mapping. Using this method, we identified 2, 4, and 3 scaffolds with high depth and breadth of coverage for strains FR13 (Figs 2 and S1), US71 (S2 and S3 Figs), and CD156 (S4 Fig), respectively. The combined size of mini-chromosome scaffolds was 1.7 Mb for FR13, 1.58 Mb for US71, and 860 kb for CD156. For FR13 and CD156 the combined length of mini-chromosome scaffolds matched the size observed on CHEF gels. However, the combined length of US71 mini-chromosome scaffolds

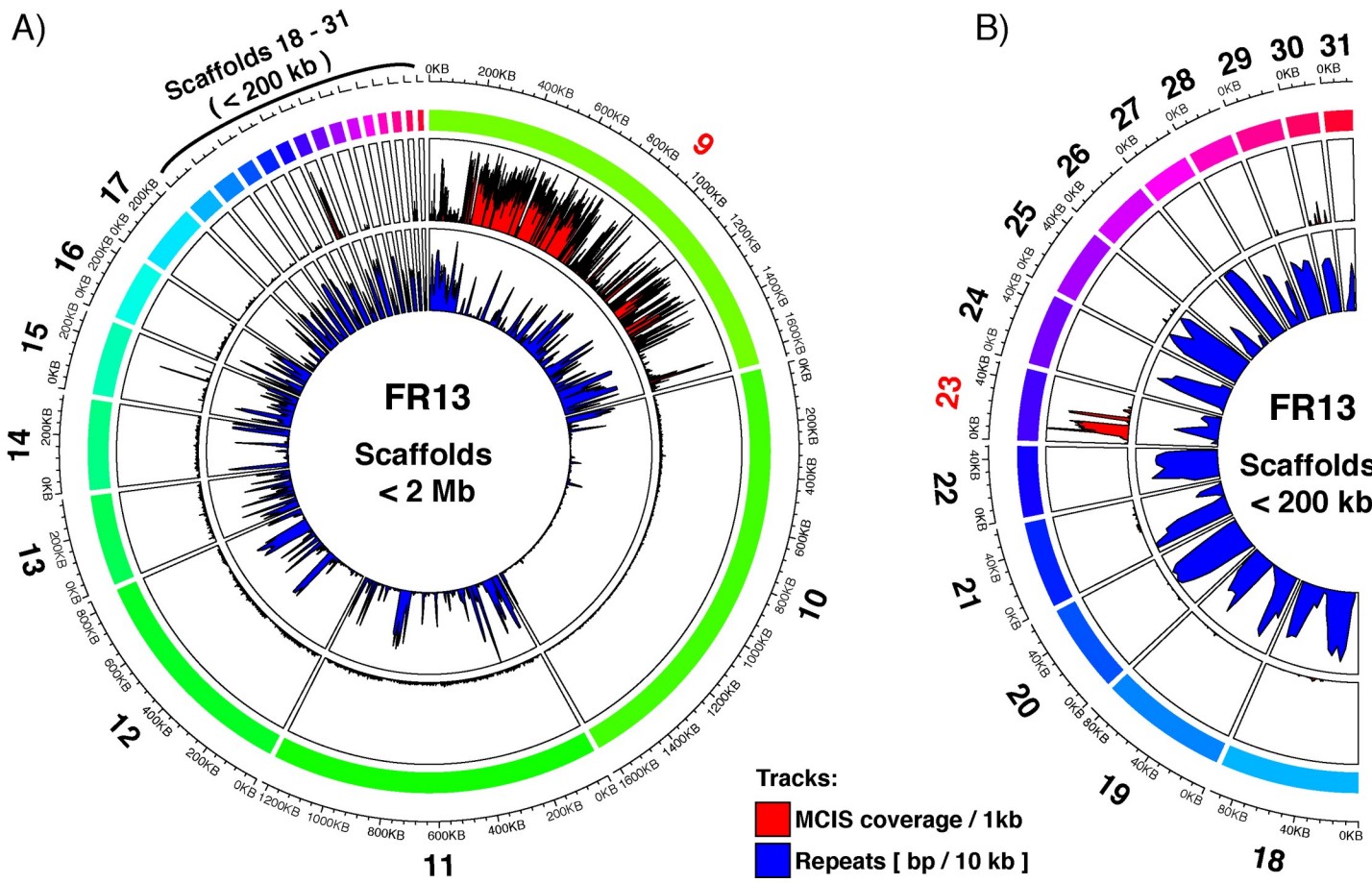

**Fig 2.** *M. oryzae* **strain FR13 contains a 1.7 Mb mini-chromosome assembled into 2 scaffolds. A)** Circos plot of mini-chromosome isolation sequencing (MCIS) read coverage and repeat content across FR13 scaffolds < 2 Mb. Outer ring (rainbow colors): FR13 scaffolds and scaffold sizes. Outer track (Red/Black): MCIS coverage per sliding window. Window size = 1000 bp; Slide distance: 500 bp. Y-axes: average coverage per 1 kb window; axis limits set to zero to maximum coverage. Inner track (Blue/Black): Repeat content per sliding window. Window size = 10 kbp; Slide distance: 5 kbp. Y-axes: repeat content in bp per 10 kb window; axis limits set to maximum. B) Circos plot of MCIS read coverage and repeat content for scaffolds <200 kb (enlarged from A).

adds up to approximately double the size of the mini-chromosome identified by CHEF analysis, indicating that US71 contains two mini-chromosomes of similar size that cannot be separated by CHEF gel electrophoresis, that there is a misassembly or that rearrangements occurred during culturing of this strain. To further investigate this, we mapped whole genome nanopore sequencing reads against the US71 genome and extracted the coverage information of mini-chromosome scaffolds (S5 Fig). Although all mini-chromosome scaffolds are overall well supported by >100x coverage, we noticed that two regions on scaffold 14 and 21, respectively, showed a drop in coverage that correlated with a drop in mini-chromosome sequencing coverage. We further compared the nanopore coverage of the mini-chromosome scaffolds with the average whole genome coverage. This analysis revealed that regions with high mini-chromosome read coverage have approximately twice as much average coverage as compared to the whole genome, whereas regions on the proposed mini-chromosome scaffolds with low mini-chromosome coverage displayed average whole genome coverage. This suggests that mini-chromosomes in US71 contain duplicated sequences that are not resolved in the current assembly.

We also noticed an enrichment of mapped mini-chromosome reads to the start of scaffold 2 in strain CD156 (S4 Fig), which might indicate that a segmental duplication of a core-

chromosomal region contributed to the emergence of the mini-chromosome in this isolate. To confirm the assembly of this region, we mapped CD156 nanopore reads against the whole genome sequence resulting in a continuous coverage of ~80x across the entire scaffold, which is consistent with the average whole genome coverage and thus confirmed the assembly (S6A Fig). This suggests that the sub-telomeric region of CD156 scaffold 2 is indeed duplicated and present on core- and mini-chromosomes. To gain further evidence for this duplication, we re-mapped the nanopore reads to the isolated scaffold 2. As expected for a duplication, we noticed a ~2x increase in coverage in this region (S6B Fig), confirming a duplication between the core genomic scaffold 2 and the mini-chromosome. We further aligned the CD156 mini-chromosome scaffolds to scaffold 2 which suggested that the mini-chromosome in this isolate represents a mosaic of repeat-rich sequences acquired from the core-genomic scaffold 2 (S6C Fig). Taken together we identified and sequenced mini-chromosomes of varying size in three isolates of *M. oryzae* that might represent isolate-specific, genomic compartments in *M. oryzae*.

## The three examined mini-chromosomes of *M. oryzae* have a different sequence composition

Based on the variation in mini-chromosome size and number, we hypothesized that they have unique sequence composition. We first investigated whether the content of mini-chromosomes is conserved in the core genome of reference strain 70–15 by globally aligning the mini-chromosomes to 70–15. This revealed that only a 761 kb fragment out of the ~1.7 Mb FR13 mini-chromosome generated an alignment matching a ~900 kb region at the end of core chromosome 2 of isolate 70–15. The mini-chromosomes of US71 and CD156 did not generate significant alignments with the reference genome.

As synteny breaks might disrupt the global alignments between the mini-chromosomes and the 70–15 genome, we further mapped mini-chromosome derived raw reads against the 70–15 assembly. The ~900 kb region on chromosome 2 of strain 70–15 showed high coverage after mapping the mini-chromosome reads of strain FR13, confirming that this region of the mini-chromosome corresponds to the end of core-chromosome 2 in 70–15. However, we did not observe unique regions with high coverage after mapping of mini-chromosome derived reads from the more distantly related strains US71 and CD156 (S7 Fig). This indicates that the mini-chromosomes of these strains contain unique sequences compared to strains FR13 and 70–15. We further examined the total amount of reads that mapped to any position in the reference genome. We found that 82.78%, 59.77%, and 55.49% of mini-chromosome reads of FR13, US71, and CD156, respectively, mapped to the same regions in the reference genome leading to local coverage peaks irrespective of the mini-chromosome reads used, indicative for unspecific mapping. However, the fraction of unmapped reads varied between FR13 (17.21%) and US71 (40.23%) or CD156 (44.51%) indicating that almost half of the mini-chromosome derived reads of US71 and CD156 originate from isolate-specific regions. Conversely, only 17% of mini-chromosome reads derived from the rice infecting isolate FR13 were strain specific, indicating higher levels of sequence conservation within the rice infecting isolates.

To further assess the uniqueness of mini-chromosomes between isolates, we performed pairwise alignments between extracted mini-chromosome scaffolds of each strain. We filtered the alignments for regions that align over > 10kb to exclude unspecific, short alignments generated by repetitive regions or local similarities. Only a small fraction of the mini-chromosomes aligned under these parameters, whereas core-chromosomes were highly similar to each other (Fig 3A). Between CD156 and US71 only 28.39% and 15.39% of the mini-chromosomes generated alignments, respectively. Between FR13 and US71 the fraction of the mini-chromosomes that generated alignments was even lower with 15.01% and 14.02% and between

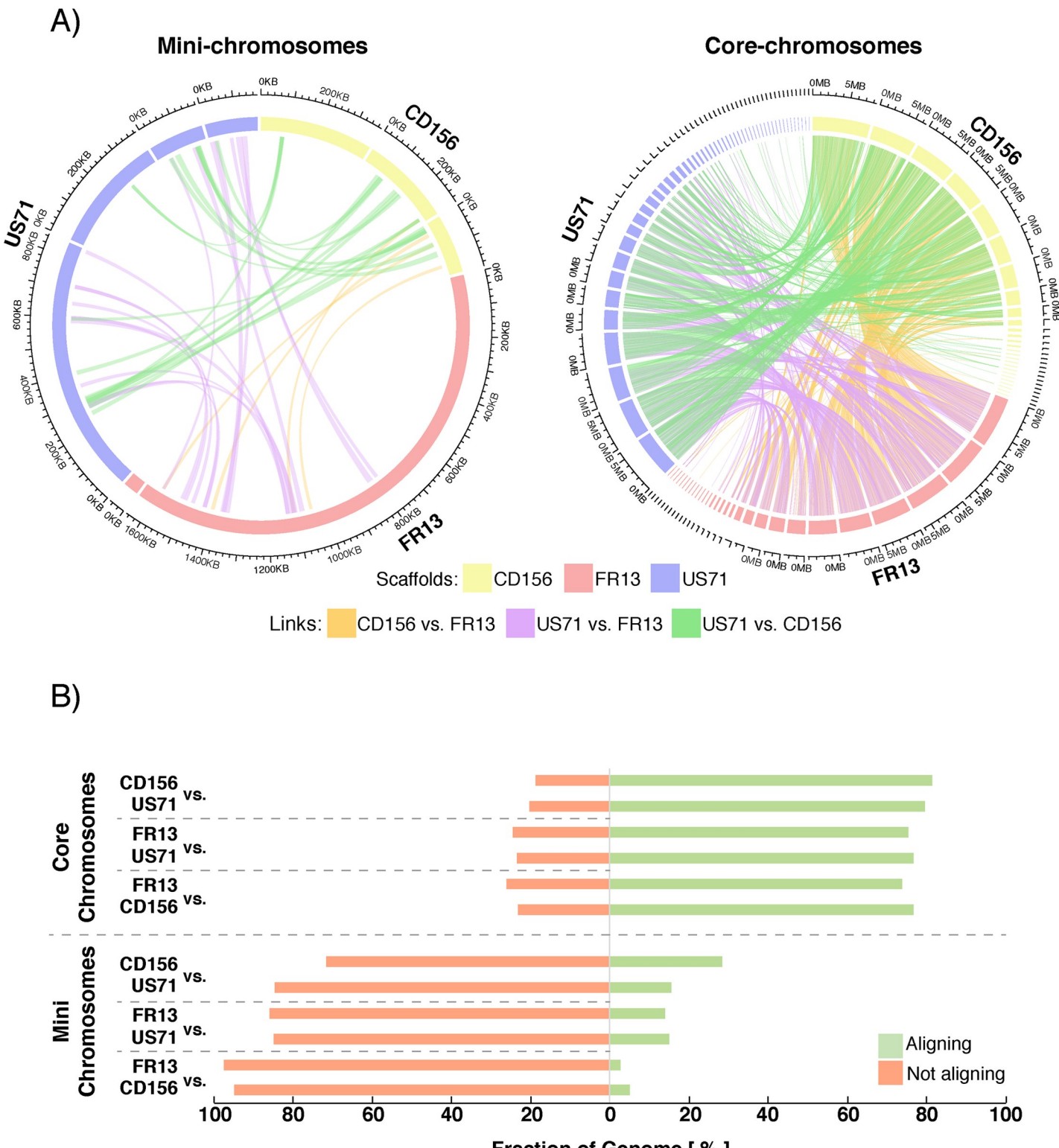

**Fig 3. Mini-chromosomes of *M. oryzae* are isolate specific. A)** Circos plots depicting alignments between mini-chromosomes (left) and core-chromosomes (right). Outer ring: Mini-chromosome scaffolds and sizes. Alignments > 10 kb are plotted as genetic links in the center. **B)** Relative fraction of mini- and core-chromosomes that generate pairwise alignments. Relative fraction of the genomic compartment that form alignments > 10 kb are shown for each pairwise alignment and each individual isolate. Approximately 75% of the core-chromosomes generate alignments under these parameters, whereas only 2.61–28.39% of mini-chromosomes do.

FR13 and CD156 it was only 5.14% or 2.61%, respectively (Fig 3B). In contrast, between 73.88% and 81.26% of the core-genome generated alignments. We conclude that the three examined mini-chromosomes of *M. oryzae* have unique sequence composition.

## The three mini-chromosomes are less conserved than core-chromosomes across *M. oryzae* lineages

To determine the extent to which the three mini-chromosome encoded sequences are conserved in various host-specific lineages of *M. oryzae* we performed whole genome alignments of 107 publicly available assemblies [41,59,60] to the mini-chromosomes of FR13, US71 and CD156. The selected assemblies are representative of 10 genetic lineages with isolates from 13 host species. We also included the *Pyricularia pennisetigena* isolate PM1 and six *M. grisea* isolates as outgroups in our analysis. As an additional control, we also aligned all genome assemblies to the core-chromosome scaffolds FR13_scaf05, US71_scaf04 and CD156_scaf05 that correspond to homologous core-chromosomes in the three isolates. We then calculated the breadth of coverage of all non-redundant alignments formed by each genome assembly with the core- and mini-chromosome scaffolds (Fig 4). Additionally, we analyzed filtered alignments that span >5 kb to exclude fragmented and repetitive alignments (Fig 4B and 4C).

This analysis showed that mini-chromosome encoded sequences are consistently less conserved than core-chromosomes across all lineages. In unfiltered alignments, mini-chromosomes yield on average between 50–80% coverage whereas core-chromosomes consistently align over >90% (Fig 4). The difference in conservation between mini- and core-chromosomes becomes even clearer when filtering for continuous alignments (>5 kb) where mini-chromosomes yield only between 10 to 50% coverage whereas core-chromosomes align over >80% (Fig 4B and 4C). Of the three analyzed mini-chromosomes, FR13 mini-chromosome sequences were the most conserved across all *M. oryzae* lineages at ~41% average breadth of coverage in contrast to ~21% and ~16% for mini-chromosome sequence of US71 and CD156, respectively (Fig 4C).

We further investigated the distribution of FR13, US71 and CD156 mini-chromosome encoded sequences across the rice lineage of *M. oryzae* using a wider set of isolates for which genome assemblies are not available [44]. In this case, we mapped reads from 131 rice infecting isolates from 21 countries that are representative of the 4 genetic lineages of the worldwide rice blast population [43,44,61] and calculated the breadth of coverage for the same mini-chromosome and core-chromosome contigs we used above. This analysis showed that the FR13 mini-chromosome encoded sequences are more conserved across rice blast isolates compared to the US71 and CD156 mini-chromosome sequences (S8 Fig). Interestingly, isolates from the recombining, presumably sexually reproducing lineage (orange lineage in [42,44]), showed higher degrees of variation in mini-chromosome sequence content.

## Mini-chromosomes have lower gene and higher repeat density than core-chromosomes

To determine the gene content of the mini chromosomes, we mapped the GEMO gene annotations [48] to our new genome assemblies using BLASTn in combination with a sequence similarity approach (see Material and Methods). Of 14515, 14013 and 14415 genes that were used as queries, 13828, 13746 and 14201 mapped to a single site in the nanopore assemblies for strains FR13, US71 and CD156, respectively.

Another 175, 179 and 44 genes mapped to multiple sites reflecting duplicated genes that were likely collapsed in the previous short-read assemblies. Taking these expansions into account, we assigned 14322, 14348 and 14304 genes to FR13, US71 and CD156, respectively.

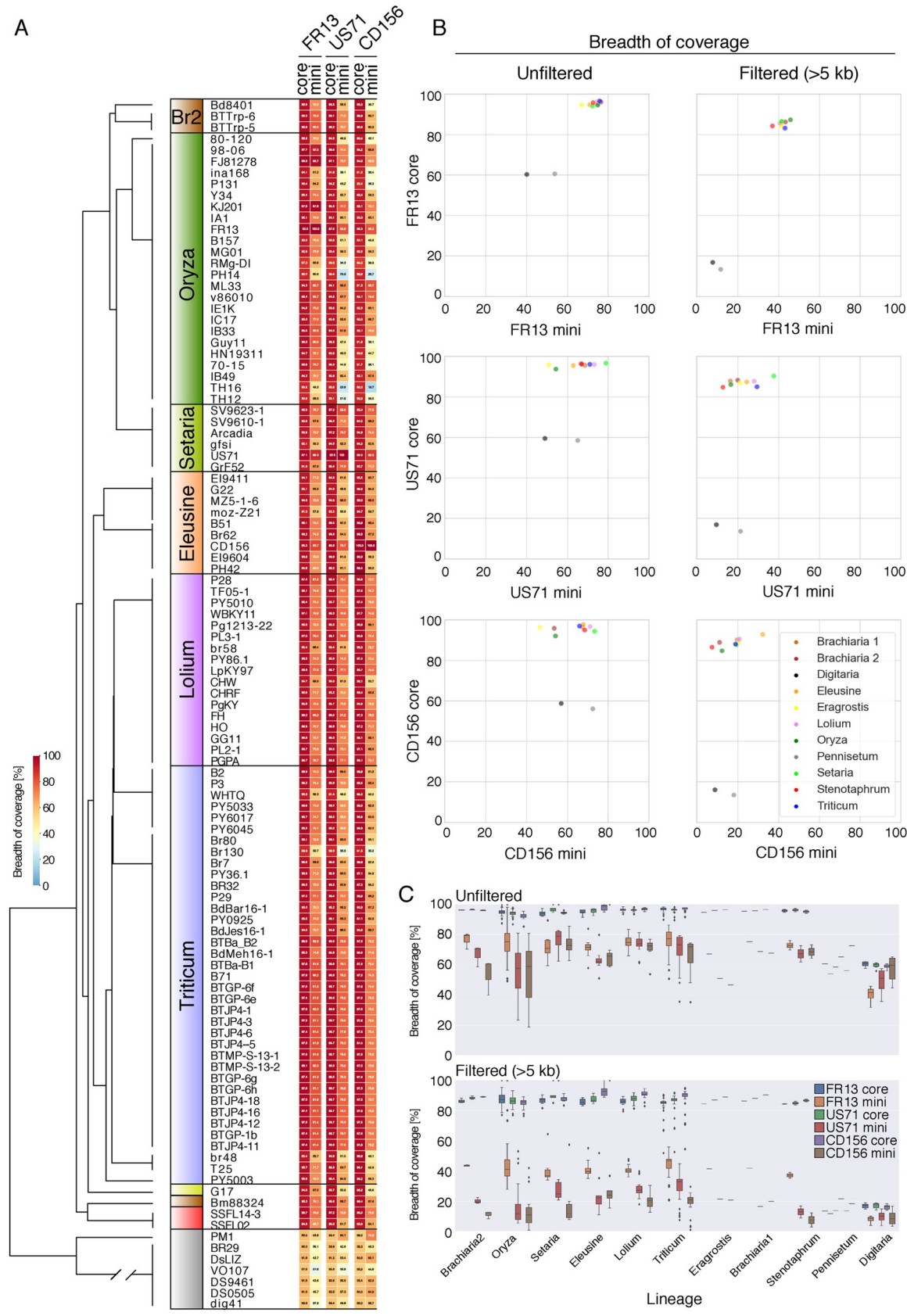

**Fig 4. Mini-chromosomes are less conserved than core-chromosomes across lineages of *M. oryzae*. A)** Conservation of core- and mini-chromosomes across host-specific lineages of *M. oryzae* based on breadth of coverage of non-redundant whole genome alignments against the three mini- and selected core-chromosome scaffolds. Heatmap values show the relative breadth of coverage of alignments for each mini-chromosome or core-chromosome scaffolds. Left panel: Schematic representation of a coalescent species tree representing host-specific genetic lineages of M. oryzae. Host genera are indicated in colors. Yellow = eragrostis lineage; Brown = brachiaria lineage 1 (isolate Bm88324) and 2 (Br2); Red = stenotaphrum lineage. **B)** Scatterplots showing the average breadth of coverage of alignments of core- over mini- chromosomes per host-specific lineage. **C)** Boxplots summarizing the breadth of coverage by genomic compartment across host-specific lineages.

Of these genes, 13963, 13985 and 14143 mapped to the core chromosomes and 359, 363 and 161 mapped to the mini chromosomes of each isolate (S1 Table). On average, in each of the three mini-chromosomes the gene content was approximately 30% lower when compared to the core genome (Fig 5A and S1 Table).

We next analyzed the repeat and GC content of the mini-chromosome scaffolds compared to the core genomes. The GC content of core- and mini-chromosomes was similar at ~50%, whereas the repeat content of the three mini-chromosomes differed from the core genome (Fig 5A and S1 Table). Indeed, the repeat content was more than twice as high in mini-chromosomes (31.38%, 27.87%, and 17.25%) compared to core-chromosomes (13.78%, 12.71% and 5.99% in strains FR13, US71, and CD156 respectively (Fig 5A).

To exclude the possibility that the observed differences are due to sample size biases that can occur by comparing mini-chromosomes that represent only small fractions of the genome to the much larger core-genome, we performed a bootstrapping analysis for each mini-chromosome scaffold (see Material and Methods). Briefly, we sampled 10,000 random fragments of the size of each mini-chromosome scaffold from the core genome and analyzed gene and repeat content. This bootstrapping approach confirmed higher repeat and lower gene content of mini-chromosomes (Fig 5B). These results indicate that even though the mini chromosomes have distinct sequences, they have common genomic features, low gene and high repeat density, that deviate from the typical composition of core chromosomes.

## Mini-chromosome encoded genes have variable patterns of conservation across lineages

Our gene content analysis resulted in a total of 42,974 genes between all three isolates. We grouped these genes into 12,900 orthogroups, of which 9,003 were conserved across all three strains and 3,897 were absent in at least one strain (S2 Table). Among the 3,897 orthogroups that were absent in at least one strain, we found 1,827 that were conserved between two strains and 2,070 were unique to single strains. We further analyzed the location of these orthogroups and categorized them into "core-genome specific", "mini-chromosome specific" and orthogroups that contain members that are present on both core- and mini-chromosomes. We found that the vast majority (99.98%) of conserved orthogroups were encoded exclusively on the core chromosomes or contained members on both core- and mini-chromosomes and only 0.02% were encoded exclusively on mini-chromosomes. Conversely, isolate-specific tribes were slightly more abundant on mini-chromosomes (S2 Table), substantiating the observation that mini-chromosomes are isolate-specific.

We further investigated the distribution of mini-chromosome encoded genes across *M. oryzae* lineages. We therefore extracted all gene models of genes that are exclusively present on mini-chromosomes and generated pairwise alignments to the 107 isolates described above using minimap2 [62]. In total, we used 305, 216, and 131 non-redundant gene models that are exclusive to the mini-chromosomes of FR13, US71, and CD156, respectively. We then extracted the best hit for each gene and filtered for completeness by applying a >90% query coverage filter. These analyses revealed variable patterns of gene conservation (both at

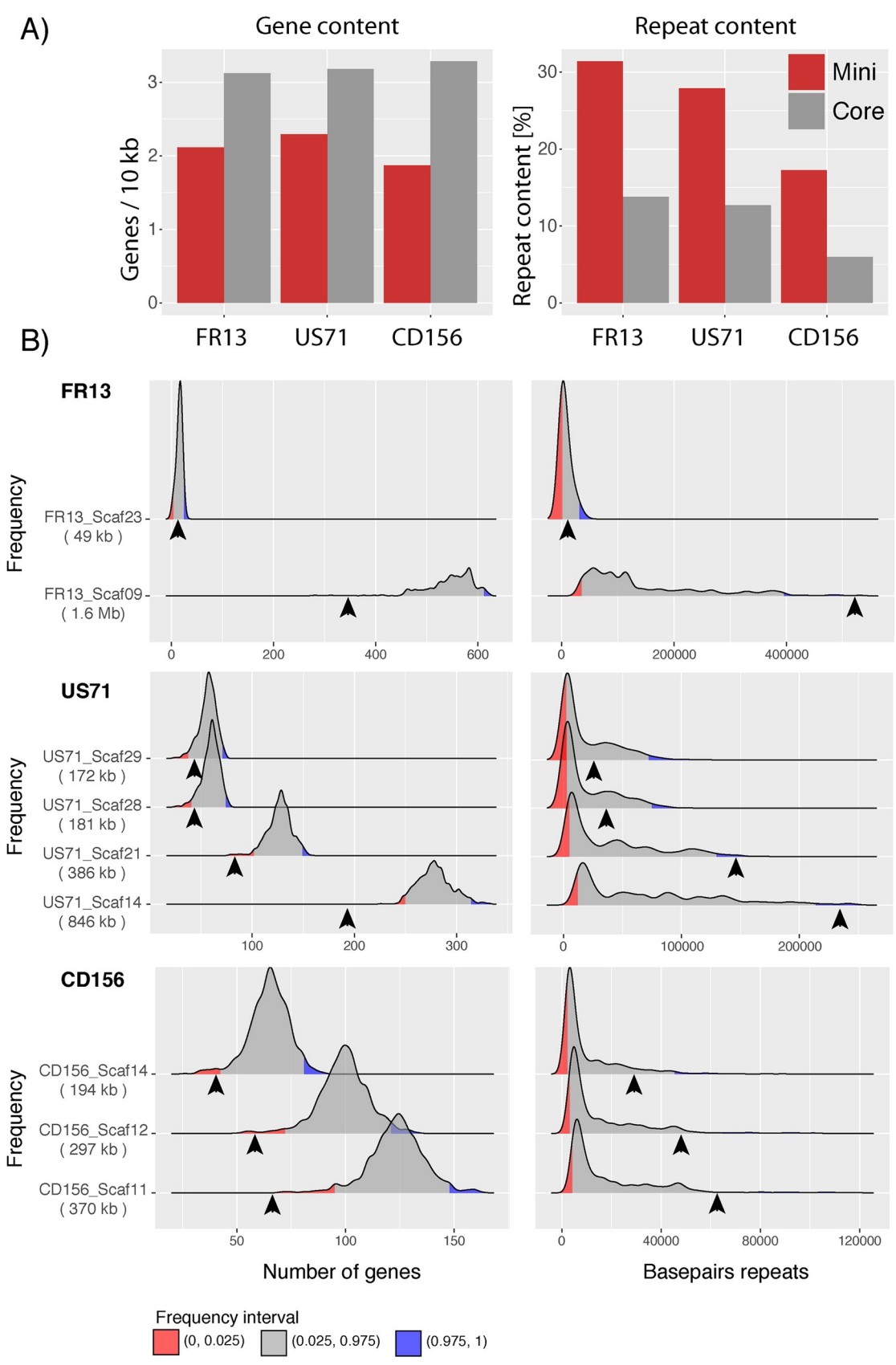

**Fig 5. Mini-chromosomes have lower gene and higher repeat content than core-chromosomes. A)** Bar plots of gene and repeat content of mini- (red) and core-chromosomes (grey) of isolates FR13, US71, and CD156. **B)** Density plots resulting from a bootstrapping analysis of gene and repeat content of isolates FR13, US71, and CD156. 10000 core-genomic fragments were randomly sampled for each mini-chromosome scaffold and distribution of genes and basepairs of repeats are shown as densities. Number of genes and repeat content of mini-chromosome scaffolds are indicated by arrowheads. The size of mini-chromosome scaffolds is given in parentheses. Lower and upper 2.5% frequency intervals are shown in red and blue. X-axis: number of genes and basepairs of repeats. Y-axis: Frequency in 10000 fragments. Y-axis limits set to min/max.

sequence level and presence/absence polymorphisms) depending on the mini-chromosome (S9, S10 and S11 Figs). FR13 mini-chromosome genes were clearly more conserved across *M. oryzae* lineages than the other two gene sets. These analyses are consistent with the patterns of mini-chromosome sequence conservation described in earlier sections (Fig 4).

## Mini-chromosomes contain highly dynamic effector and virulence-related loci

To determine functional categories of mini-chromosome encoded genes, we performed a Hidden Markov Model (HMM) scan against the Pfam database (https://pfam.xfam.org/) using a total of 772 non-redundant, mini-chromosome encoded proteins. More than 50% of these proteins (415) did not contain any known domain, whereas 357 proteins contained Pfam domains. Certain domains were shared between at least two mini-chromosomes and included transcription factor DNA-binding domains, especially Zn(2)-Cys(6) zinc-finger, DDE-superfamily endonuclease, Tc5 transposase DNA-binding, and Methyl-transferase domains as well as domains of unknown function, amino-acid permease, protein kinase, and glycosyl-hydrolase domains. However, the relative abundance of these domains varied between individual isolates (S3 Table) and it is unclear if the presence of these protein domains on the mini-chromosomes has functional implications.

The most abundant, isolate-specific domains on mini-chromosomes were present in the rice isolate FR13 and included cytochrome P450 and polyketide-synthase domains. Both of these domains are involved in pathogenicity in *M. oryzae* and other plant pathogens [17,63]. Interestingly, we identified a polyketide synthase as Ace1 (avirulence conferring enzyme 1), which is part of a large secondary metabolite cluster and triggers an immune response in host plants carrying the resistance gene Pi33 [63].

Virulence and host adaptation of *M. oryzae* are largely determined by genes encoding secreted effector proteins. To identify the degree to which secreted protein genes occur on mini-chromosomes, we predicted the secretome of all isolates using signalp 2.1 in combination with targetp and tmhmm to remove mitochondrial and transmembrane proteins (see Material and Methods). This resulted in 1,394, 1,558, and 1,611 secreted proteins in strains FR13, US71, and CD156, respectively. We found 50, 8, and 2 secreted proteins encoded on mini-chromosomes, most of which correspond to uncharacterized hypothetical proteins (S4 Table).

We further investigated whether candidate effector genes are present on the mini- and core-chromosomes using 27 characterized effectors and 167 predicted MAX-effectors (Magnaporthe AVRs and ToxB) identified by de Guillen *et al.* [64]. MAX-effectors represent a unique class of proteins that is expanded in *M. oryzae* and contain proteins that are sequence unrelated but share a similar core structural fold. Using tblastn, we identified 75, 67, and 63 proteins with similarity to known or predicted MAX-effectors in the genomes of FR13, US71, or CD156, respectively. Most of these genes were located on the core-chromosomes (S12 Fig). However, we found 5 genes in FR13 and 2 genes in US71 that were located on the mini-chromosomes. Interestingly, 3 of these genes were duplicated either on the same mini-chromosome or between the mini- and core chromosomes, suggesting structural, genomic rearrangements following segmental duplication events. Among the duplicated effector genes,

we found the rice isolate specific effector AVR-Pik on the mini-chromosome of FR13 (S12A Fig). Interestingly, we found two genes encoding different variants, AVR-PikD and AVR-PikA, located towards both ends of the FR13 mini-chromosome scaffolds (S12A Fig). This is consistent with previous observations by Kusaba *et al.* [55], who identified the same effector variants on a 1.6 Mb mini-chromosome in the Japanese isolate 84R-62B and might indicate that certain mini-chromosomes are maintained in *M. oryzae* populations.

### Patterns of genomic rearrangements around effector and virulence-related loci in mini-chromosomes

Our Pfam and effector analyses suggested that some virulence related genes, including AVR-Pik and Ace1, are located on mini-chromosomes and that these regions undergo rearrangements that possibly involve inter-chromosomal translocation events. To explore this further, we extracted the AVR-Pik and Ace1 loci from other highly contiguous genome assemblies of the related strains 70–15, FJ81278, and Guy11 and analyzed the macro-synteny of corresponding regions for signs of genomic rearrangements. AVR-Pik is present in the reference strain 70–15 and is encoded in the sub-telomeric region at the end of chromosome 2 (supercontig_8.2) and on contigs 13 and 21 in strains Guy11 and FJ81278, respectively. In strain 70–15 the AVR-Pik gene resides in the region that is syntenic to the 761 kb region on the mini-chromosome of FR13 described earlier. However, comparison of the corresponding region of Guy11 suggested large scale rearrangements and variable degrees of synteny around the AVR-Pik locus in different rice isolates (Fig 6). Additionally, we observed that the syntenic regions that are shared between the analyzed strains encode different variants of the AVR-Pik effector. Whereas the first syntenic region of the FR13 mini-chromosome encodes the variant AVR-PikA, syntenic regions in 70–15 and Guy11 encode the variant AVR-PikC. Furthermore, we identified a 82 kb region in isolate FJ81278 encoding AVR-PikD with high sequence similarity and synteny to the FR13 mini-chromosome scaffold 23 that encodes the same AVR-Pik variant, suggesting that the two copies of AVR-Pik on the mini-chromosome originated from independent genomic locations and recombined on the mini-chromosome.

The polyketide synthase Ace1 is also located on the FR13 mini-chromosome. Ace1 is part of a large secondary metabolite cluster that spans a region of approximately 70 kb [63]. To analyze the genomic context around this cluster on the mini-chromosome, we extracted the gene sequences of the whole cluster in FR13 and analyzed syntenic regions in the aforementioned isolates as well as US71 and CD156. In rice isolates and the foxtail millet isolate US71, the Ace1 cluster and the order of the macrosyntenic region are well conserved (Fig 6B). Interestingly, the ACE1 cluster is encoded on core-chromosomes in strains 70–15, Guy11, and US71, based on the size of the contigs, suggesting inter-chromosomal rearrangements that involve core- and mini-chromosomes. In the more distantly related strain CD156, the Ace1 cluster is less conserved and the syntenic region only spans across 195 kb, reflecting the genetic diversity and possibly genomic rearrangements between rice isolates and distantly related lineages. Strikingly, the macrosyntenic region on the FR13 mini-chromosome is disrupted by a 392 kb insertion after the Ace1 cluster that is absent in the core-chromosomes of other isolates, further substantiating the hypothesis that mini-chromosomes undergo complex genomic rearrangements and might acquire isolate specific sequences.

### Core-genomic rearrangements are associated with mini-chromosome emergence in *M. oryzae*

To further investigate the possible role of core-genomic rearrangements in mini-chromosome emergence, we compared the genome structure of all isolates to closely related reference

A)

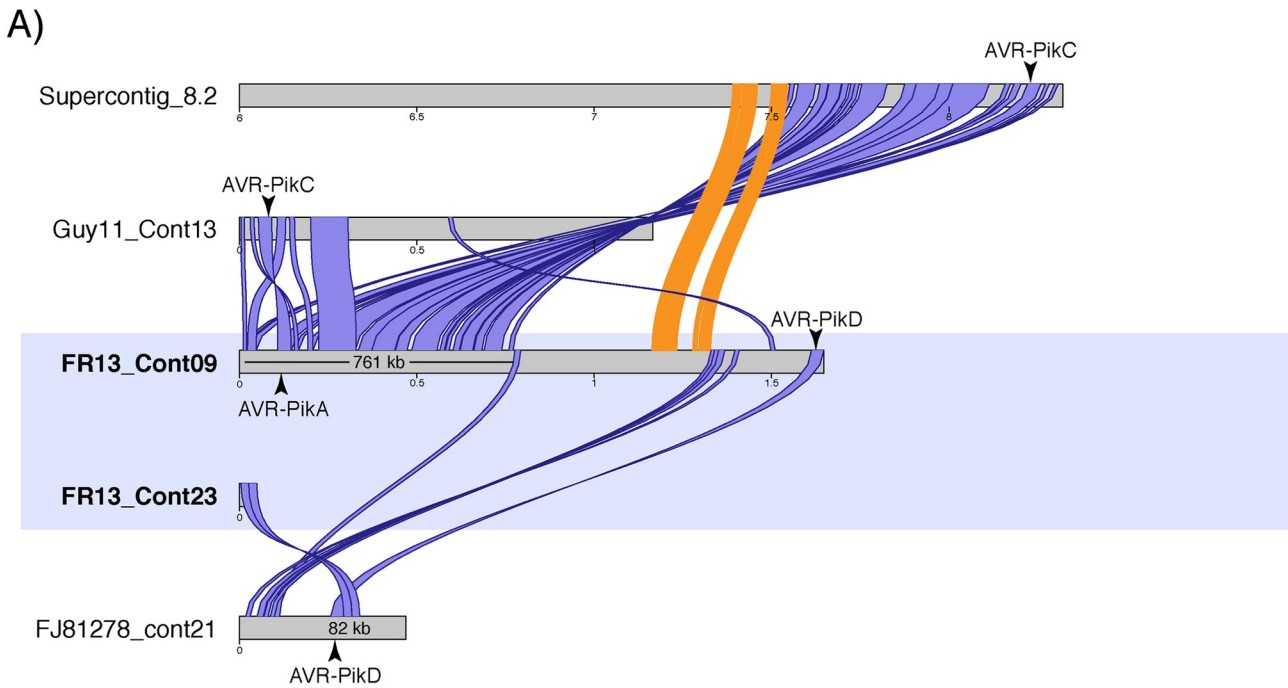

B)

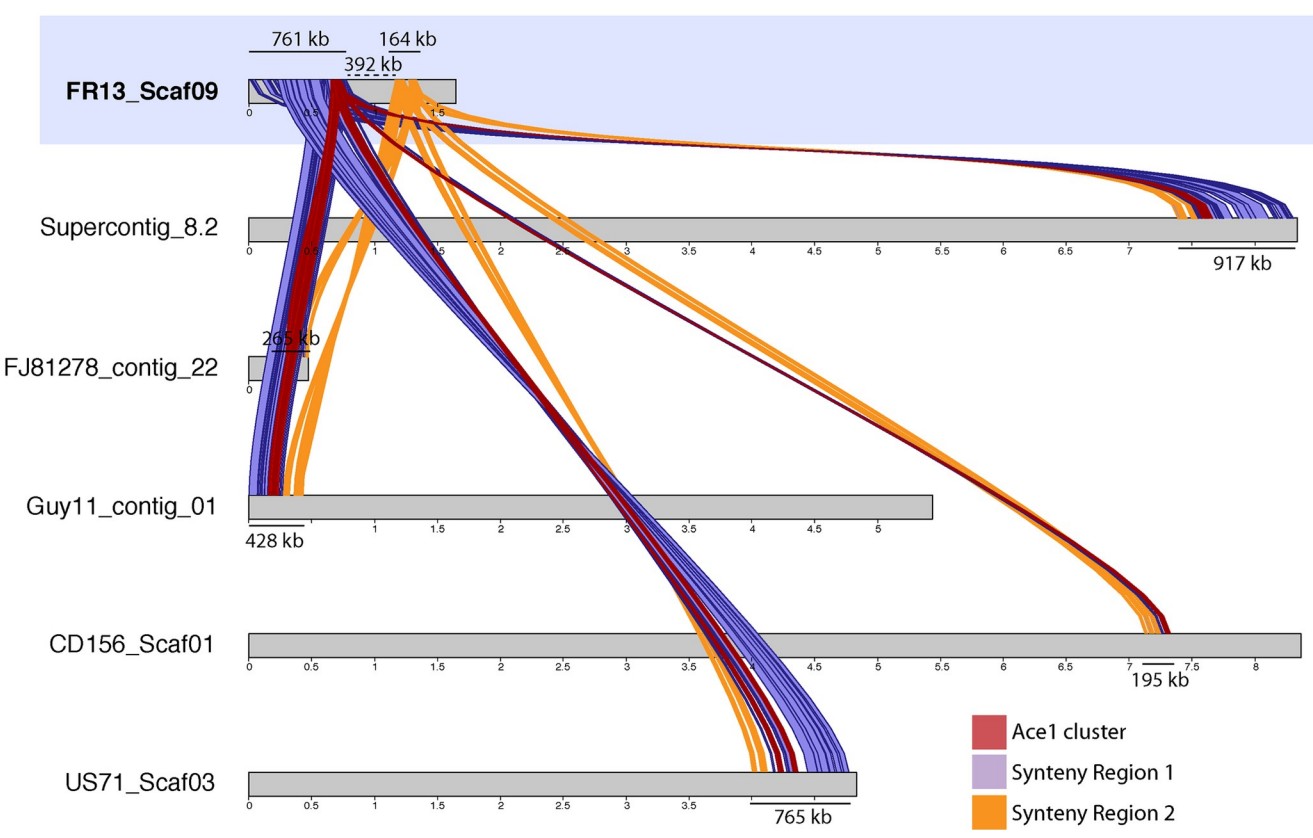

**Fig 6. Genomic rearrangements around virulence-related loci on mini-chromosomes. A)** Synteny analysis around the AVR-Pik loci on the FR13 mini-chromosome and corresponding chromosomes and contigs in the related isolates Guy11, FJ81278, and the reference strain 70–15. The 761 kb region around the variant AVR-PikA on the mini-chromosome with high similarity to the end of chromosome 2 of the reference strain 70–15 is shown in purple. The makrosyntenic region between 70–15 and FR13 is disrupted on the mini-chromosome (synteny break between region 1 and 2). The synteny is disrupted earlier in Guy11 and there are 2 inversions around the AVR-Pik locus. Another syntenic region (82 kb) is present between the end of the mini-chromosome encoding the AVR-PikD variant and contig 21 of isolate FJ81278. Syntenic regions larger than 10 kb are shown. Illustration of supercontig 8.2 starts at 6 Mb for better visualization. **B)** Synteny analysis around the Ace1-containing secondary metabolite cluster on the FR13 mini-chromosome and corresponding contigs of isolates Guy11, FJ81278, 70–15, US71, and CD156. The Ace1 cluster is located at the end of the 761 kb syntenic region between the FR13 mini-chromosome and supercontig 8.2 of isolate 70–15. The synteny around the Ace1 locus is highly conserved in closely related isolates, including the foxtail millet isolate US71. In strains Guy11, US71, and 70–15 the cluster is located on core-chromosomes. The macro-synteny around the Ace1 cluster is disrupted by a 392 kb insertion on the FR13 mini-chromosome. The Ace1 cluster is shown in red and the two syntenic regions (761 kb and 164 kb) are shown in purple and yellow, respectively. Mini-chromosome scaffolds are highlighted in blue.

genomes. In the case of FR13 and US71, the most closely related reference genome is 70–15. For CD156, we compared the genome structure to the chromosome quality reference assembly of the eleusine isolate MZ5-6-1 [65] in addition to 70–15. We identified several isolate-specific rearrangements in form of intra- and inter-chromosomal inversions and translocations (S13, S14 and S15 Figs). Most strikingly, we identified a major genomic rearrangement in the form of an intra-chromosomal inversion/translocation between core-chromosome 2 in 70–15 and scaffold 1 in FR13 that was directly adjacent to the syntenic region we identified between chromosome 2 and the mini-chromosome of FR13 (Figs 7A and S13) that was well supported by >100x nanopore read coverage (S16 Fig).

This association between a major core-genomic rearrangement and the split between core- and mini-chromosome in FR13 suggests that mini-chromosomes emerge through structural changes in the core genome. We further analyzed the gene-, repeat- and GC-content in the regions surrounding this rearrangement and found that blocks of repeat rich regions are associated with the observed rearrangements (Fig 7B), suggesting a role of repeats in facilitating both, core-genomic structural changes and emergence of mini-chromosomes. This finding is further supported by the observed duplication of the repeat-rich region of the core-genomic scaffold 2 in CD156 that is duplicated on the mini-chromosome (S4 and S6 Figs).

We also took advantage of the availability of high-quality genome assemblies of isolates 70–15, MZ5-1-6 and BR32, which do not carry mini-chromosomes, to further investigate the relationship between mini-chromosomes and core-chromosomes. We analyzed unfiltered alignments of the three mini-chromosomes to genome assemblies of isolates 70–15 (rice infecting lineage), MZ5-1-6 (eleusine infecting lineage) and BR32 (wheat infecting lineage). This analysis revealed that the FR13 mini-chromosome is the only one to form continuous alignments with core chromosome sequences (chromosome 2) of each of the tested isolates (S17 Fig). Conversely, US71 and CD156 mini-chromosome sequences only generated fragmented alignments of repetitive nature (S17 Fig).

## *M. oryzae* mini-chromosomes contain canonical telomeres

Our observations along with the results obtained with the isolate B71 by Peng et al., 2019 [56] imply that sub-telomeric regions associated with rearrangements could contribute to mini-chromosome emergence. Therefore, we investigated the extent to which mini-chromosomes contain canonical telomeric repeats. We searched for canonical telomeric repeats in our assemblies after reordering the contigs according to the genomic structure of the reference genomes 70–15 and MZ5-6-1 using MAUVE [66]. We identified telomeric repeats in the isolates FR13 and US71, but not in CD156. In US71, we identified 10 scaffolds with telomeric repeats on one end of the scaffold. Co-linearity analysis to 70–15 placed these telomeres at one end of each core chromosome and to 3 small scaffolds that did not align to 70–15, including

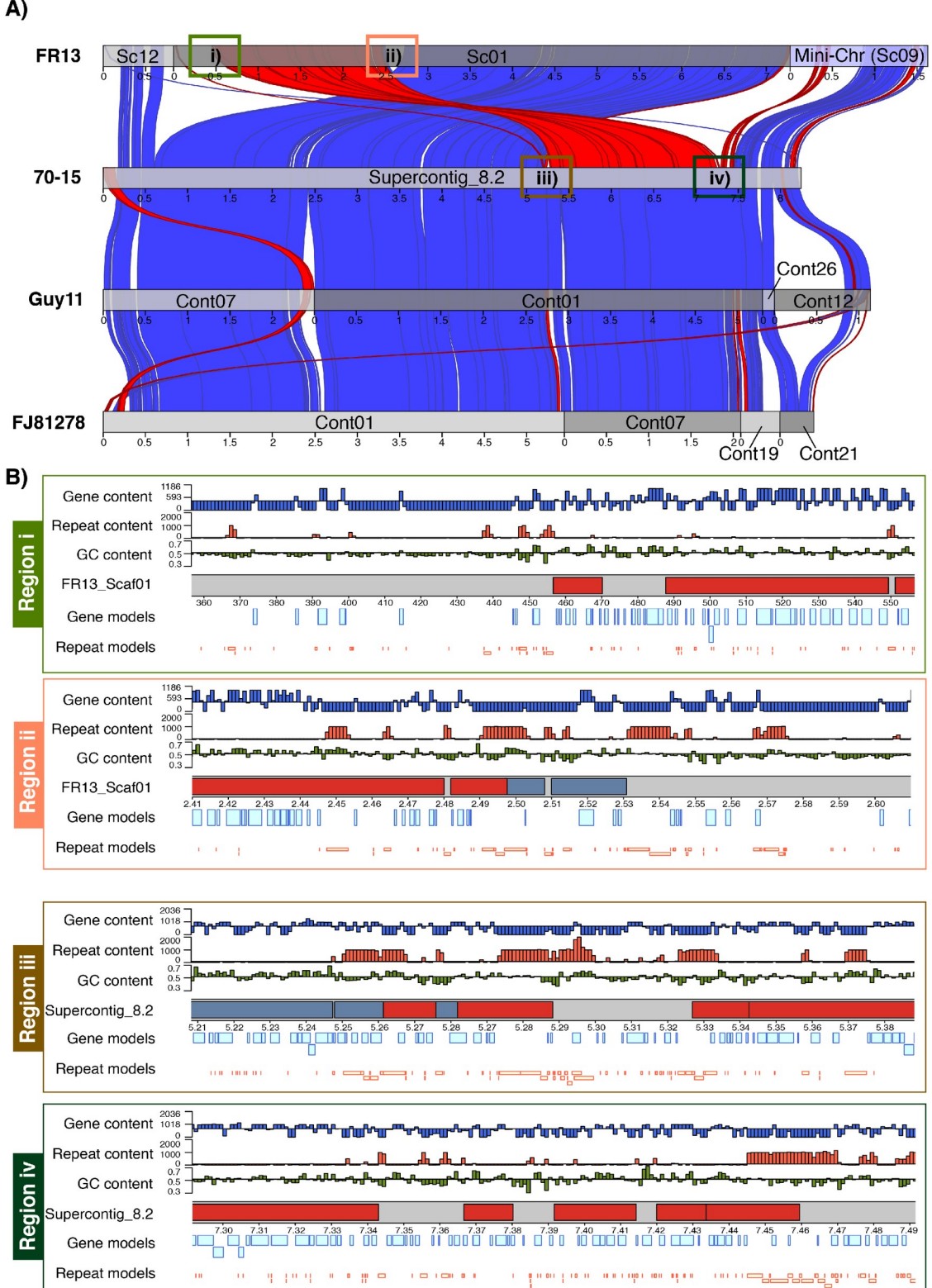

**Fig 7. Core-genomic rearrangements are associated with mini-chromosome emergence. A)** Synteny analysis of 70–15 chromosome 2 and syntenic scaffolds and contigs in FR13, Guy11, and FJ81278. Syntenic regions between isolates are shown as genetic links in blue (alignments in forward direction) and red (reverse direction). Aligning regions are indicated as red and blue boxes. Scaffold and contig breaks are indicated as grey tones. The mini-chromosome scaffold 9 is shown in bright blue. Regions

around the major rearrangement are numbered and indicated by colored boxes. **B)** Analysis of GC-content, gene-content, and repeat-content in regions i-iv surrounding the major rearrangement. Colors correspond to colors in (A). Axis limits for GC-content was set to 0.5–0.7. Axis limits for gene- and repeat-content was set to 0 / max and the center line represents the average of the contig. Gene- and repeat content is plotted as bp per 1kb sliding window. Gene and repeat models are plotted below in bright blue and bright red, respectively.

the mini-chromosome scaffold 21 (S14 Fig). We further identified 13 telomeres in isolate FR13, including two scaffolds that resemble complete chromosomes (S13 Fig). Interestingly, we found classical telomeric repeats at the end of both mini-chromosome scaffolds, indicating that mini-chromosomes are linear and that telomere and sub-telomere associated repeats are associated with mini-chromosome emergence.

## Discussion

Genomes of eukaryotic plant pathogens are notorious for being highly dynamic. Chromosome scale structural variation, including supernumerary mini-chromosomes, has been documented by electrophoretic karyotyping [14] but has not been studied extensively at the sequence level mainly due to methodological limitations. Here, we took advantage of recent technical developments to reassess the genomes of 4 previously sequenced, host-specific isolates of the blast fungus *M. oryzae*. We used a combination of long- and short-read sequence data to generate near chromosome quality genome assemblies. We found that supernumerary mini-chromosomes are common in isolates of host-specific lineages of the blast fungus *M. oryzae*. We then used a technique we termed mini-chromosome isolation sequencing to collect short-read data from isolated mini-chromosomes and match them to scaffolds in the whole genome assemblies. We detected single mini-chromosomes in the rice and goosegrass isolates, and possibly two mini-chromosomes in the foxtail millet isolate of *M. oryzae*. Our study validates, complements and extends the findings of Peng et al. [56], which described mini-chromosomes in the wheat blast lineages of *M. oryzae*. Therefore, all major lineages of this pathogen carry mini-chromosomes and future genomics studies of the blast fungus pathogen will need to take this into account.

Comparative analyses of the mini-chromosomes sequences revealed that they are highly variable between isolates indicating that they may have emerged independently or that they have diverged rapidly after an ancient emergence event. Comparative analyses with core chromosomes indicate that the mini-chromosomes partially originate from rearrangements and segmental duplications of core-chromosomes. However, the mini-chromosomes share common features such as low gene and high repeat density. We also located several genes encoding secreted proteins on mini-chromosomes although the numbers varied between the three isolates. Due to the small sample size used in this study, we cannot at this point draw strong conclusions from these observations. However, our data suggest that mini-chromosomes emerge from core chromosome ends and the gene and repeat composition of the mini-chromosomes appears to reflect this origin.

We could not determine the precise structure of the US71 mini-chromosomes. Our results indicate that US71 mini-chromosome scaffolds contain at least two regions that are duplicated in the genome that were not resolved in the current assembly. It is generally challenging to assemble duplicated regions due to several technical reasons. It is possible that duplicated regions that are present on core- and mini-chromosomes in this isolate are simply hard to assemble separately. It is also possible that genome rearrangements have emerged during culturing and interfered with matching mini-chromosome reads to scaffolds. In the future, technical advances are needed to address this type of issues. For example, improving MCIS by

including long-range sequencing of purified mini-chromosomes would lead to better mini-chromosome assemblies.

We located the genes for the effector AVR-Pik and the polyketide synthase Ace1 to the mini-chromosome of the rice-infecting isolate FR13 even though these genes are located on core-chromosomes in other isolates. Analyses of macrosynteny around these regions showed that mini-chromosomes undergo large-scale rearrangements, and thus, could contribute to genomic plasticity within *M. oryzae* populations. Additionally, we provide evidence that mini-chromosomes are associated with extensive rearrangements of core-chromosomes. Overall, our results demonstrate the value of mini-chromosome isolation sequencing (MCIS) as a scalable method to reliably identify mini-chromosomes in whole genome assemblies. This will allow us to study the biogenesis and biological relevance of this unique genomic compartment.

Our study, along with Peng et al. [56], reveals that mini-chromosomes are present in independent lineages of *M. oryzae*. Although we found mini-chromosomes in the examined rice, foxtail millet and goosegrass isolates of *M. oryzae*, we didn't detect any in the wheat blast isolate BR32 (Fig 1), an isolate that traces back to early outbreaks of wheat blast in 1991. This is consistent with the recent study by Peng *et al.* [56], which documented mini-chromosomes in the wheat blast isolates B71 and P3 collected in 2012 but not in T25 collected in 1988. It is possible that wheat blast isolates collected soon after the emergence of this disease in the 1980s lack mini-chromosomes. The lack of meiotically unstable mini-chromosomes in these isolates may be due to sexual reproduction, which was common among isolates from the early phases of the epidemic [67]. Meiotic instability could also explain the lower degree of conservation of the FR13 mini-chromosome in the recombining lineage of rice blast isolates (S8). However, we cannot rule out the possibility that these isolates have lost their mini-chromosomes over time due to long term culturing under laboratory conditions. Similar loss of accessory chromosomes within weeks has been demonstrated *in vitro* for *Z. tritici* [20]. The observed variation in mini-chromosome content is consistent with earlier studies showing that the occurrence of mini-chromosomes is variable within and between lineages of *M. oryzae* [52,53].

The mini-chromosomes we studied share little sequence similarity with each other indicating that they are not conserved across host specific isolates but rather emerged independently or diversified more rapidly than core-chromosomes throughout lineage diversification of *M. oryzae* (Fig 3). In contrast, mapping of mini-chromosome derived reads of wheat isolate P3 to the B71 mini-chromosome sequence revealed several overlapping regions [56]. Despite the overlaps, these regions seem to be interspersed with sequences that are specific to each individual mini-chromosome. This is consistent with the view that mini-chromosomes can emerge frequently from rearrangements of core-chromosomes. Indeed, we detected significant similarity in whole genome alignments to the mini-chromosomes in isolates from several lineages (Fig 4). However, it is unclear to which extent the aligned sequences correspond to mini-chromosomes or to matching regions from core-chromosomes.

Our comparison of the FR13 mini-chromosome to high quality reference assemblies indicates that mini-chromosomal sequences match core chromosome regions (S17, S13 and S15 Figs). Interestingly, the matching core chromosome regions varied depending on the isolate. Whereas FR13 mini-chromosome matched the end of chromosome 2 in rice isolate 70–15, the eleusine isolate MZ5-1-6, and the triticum isolate BR32 (S17 Fig), it matched chromosome 1 (scaffold 08) in the eleusine isolate CD156 (S15 Fig). These observations are consistent with a model where mini-chromosomes and core chromosomes exchange sequences. However, more fine-tuned comparisons of closely related sibling isolates are necessary to refine this model and determine the frequency at which such exchanges occur. Future comparative analyses, notably mini-chromosome isolation sequencing of natural populations of *M. oryzae*, will shed light on the mechanisms of mini-chromosome emergence, maintenance and evolution.

Despite their independent origin, mini-chromosomes share common genetic features. We notice that elevated repeat content and lower gene densities are consistent features of mini-chromosomes (Fig 5) [56]. This is surprising given that all mini-chromosomes share overall little sequence similarity and seem to have emerged independently. What is the underlying mechanism that facilitates emergence of sequence unrelated mini-chromosomes with common genomic features? Our results suggest that mini-chromosomes might emerge from core genomic rearrangements involving primarily core-chromosome ends (Figs 6, 7 and S4). These regions are known to have higher repeat content and to be more dynamic than central regions of chromosomes [11,56,68]. It seems plausible that repeats are involved in mini-chromosome formation, a notion that is further supported by our observation that core genomic rearrangements and mini-chromosome emergence are associated with blocks of repetitive sequences (Fig 7B). Repeats facilitate genomic rearrangements, such as inter-chromosomal translocations, in several plant pathogenic fungi [6,11,20,69,70]. Under stress conditions, specific classes of transposable elements can get activated and induce genomic rearrangements [71]. Interestingly, Peng et al. [56] observed that the B71 mini-chromosome contains more active repeats than core chromosomes based on lower rates of repeat-induced point (RIP) mutations. The emerging model is that transposon activation generates genomic instability, primarily at core-chromosome ends, resulting in the genesis of mini-chromosomes. Future comparative analyses of mini-chromosomes from closely related isolates, ideally within the same clonal lineage, will further reveal the precise genetic elements associated with the emergence and divergence of mini-chromosomes.

What are the implications of mini-chromosomes for adaptive evolution of the blast fungus? Our results, together with other's suggest that mini-chromosomes form a gene poor, repeat-rich genomic compartment that contributes to genome rearrangements and gene flow [11,55,56]. Given that effector genes are associated with mini-chromosomes in *M. oryzae*, this genomic architecture is another example of the two-speed genome concept in which particular compartments contribute to adaptive evolution. This could happen through several mechanisms. Mini-chromosomes may represent an intermediate stage of large rearrangements that facilitate generation of structural variations across the genome and might contribute to divergence and gene gains/losses of virulence related genes.

One hint about the adaptive nature of mini-chromosomes is the presence of AVR effector genes. Indeed, the FR13 1.7 Mb mini-chromosome has a similar size and architecture to a previously reported mini-chromosome of another rice-infecting isolate, 84R-62B [55]. Both FR13 and 84R-62B mini-chromosomes carry the AVR-PikA and AVR-PikD variants of the effector AVR-Pik. Kusaba et al. [55] demonstrated that the presence of each of the two effector variants on the 84R-62B mini-chromosome is associated with allele specific recognition in rice cultivars carrying the cognate variants of the immune receptor Pik. They further suggested that rearrangements between the mini- and core chromosomes, as well as mini-chromosome loss can lead to AVR-Pik presence/absence polymorphisms that determine virulence on Pik rice plants, thereby clearly demonstrating that genes on mini-chromosomes can impact virulence. Another effector, AVR-Pita, is also encoded on mini-chromosomes in various rice blast isolates and multiple independent rearrangement events are thought to have generated AVR-Pita containing mini-chromosomes [11].

A species such as *M. oryzae* that consists of genetically defined host-specific lineages may benefit from a process that facilitates horizontal transfer of genetic material. Genetically diverse isolates that infect a common host may give rise to new variants that increase the diversity and adaptive potential of local blast populations [72]. Future analyses are needed to determine the extent to which transfer of supernumerary chromosomes drives the evolution of *M. oryzae* as reported for other plant pathogenic fungi [19,27,55,73].

In conclusion, it is of utter importance to understand the biology of supernumerary mini-chromosomes and its impact on genomic diversity and gene flow. Our study lays the basis for studying the role of mini-chromosomes in defining the genetic identity of host specific lineages of *M. oryzae*. In the future we will integrate mini-chromosome sequencing into pangenome studies of *M. oryzae* to further our understanding on the genetic events that shape the evolution of host-specific lineages of *M. oryzae*.

## Material and methods

### Biological material

The *M. oryzae* isolates analyzed in this study were field isolates collected from different hosts and different regions and represent four host-adapted lineages. FR13 was isolated from japonica rice in France in 1988, US71 was isolated from *Setaria sp*. (foxtail millet) in the USA, CD156 was isolated from *Eleusine indica* (goosegrass) in Ferkessedougou, Ivory Coast in 1989, and BR32 was isolated from wheat in Brazil in 1991. All isolates were acquired from Elisabeth Fournier and have been previously reported in the GEMO project [48].

### DNA extraction for whole genome sequencing and mini-chromosome isolation sequencing

For mycelium propagation and whole genome sequencing, *M. oryzae* isolates were cultured on complete medium agar (CM). High molecular weight genomic DNA from *M. oryzae* was extracted from mycelia of 7-days old cultures following the method described in [74]. Genomic DNA was quantified on a TapeStation (Agilent) and treated with DNAse-free RNAse. RNAse treated DNA was sheared using either a gTUBE or a 22 Gauge needle. Sheared DNA was captured using AMPure beads (Beckman Coulter, Indianapolis, US) and eluted in 45 μl water.

For mini-chromosome isolation sequencing, 8 blocks of 7-days old mycelium were transferred into 150 ml YG-medium (5g/L yeast extract, 20g/L glucose) and incubated at 24°C and 120 rpm for 3 days. Mycelium was harvested by filtering the culture through two layers of miracloth. Protoplasts were generated by incubation in sterile *Trichoderma harzianum* lysing enzymes solution (150 mg lysing enzymes in 15 mL 1 M Sorbitol) for 2–4 h. Protoplasts were harvested by filtering through two layers of miracloth followed by centrifugation for 10 min at 1,500 rpm and washed twice in 1 M Sorbitol. Quality of protoplasts was observed microscopically. Protoplasts were then resuspended in 100–200 μl 1 M Sorbitol / 50 mM EDTA, embedded in 2x volumes of 1% certified megabase agarose (Biorad) and incubated in proteinase K containing NDS buffer (10 mg/mL laurylsarcosine, 100 mM TRIS-HCl pH9.5, 500 mM EDTA, proteinase K 200 μg/ml) at 50°C for 48 h. Plugs were then washed three times with fresh 50 mM EDTA for 1 h prior to CHEF gel electrophoresis (detailed protocol available on protocols.io, [75]).

### Whole genome sequencing and assembly

Libraries for whole genome sequencing were prepared by following the 1D protocol from Oxford Nanopore. Sequencing runs were performed using MinION R9.4 (Oxford Nanopore Technologies, Oxford, UK). Sequence reads were assembled into contigs using Canu (v1.6 and v1.7) [76]. Contiguity was further improved by merging highly similar contig ends with identity >98% and alignment length >10 kb (for three contigs the length of the alignment was between 8 kb and 10 kb) after whole genome alignments using the nucmer application of the MUMMER3 package [77] and SSPACE [78]. After extracting and merging matching contig

ends, we tested for read support for both ends and reads spanning the entire region (S5 Table). We then improved base calling quality of the assemblies by applying a polishing pipeline (https://github.com/nanoporetech/ont-assembly-polish) consisting of two iterations of racon (https://github.com/isovic/racon; v1.3.2) and two iterations of pilon v1.22 [79] polishing using Nanopore and published Illumina reads (Illumina reads from [48]). To assess the performance of the polishing pipeline and the overall quality of the genome assemblies, we performed a benchmarking universal single-copy orthologs (BUSCO) [80] analysis using the Sordariomycota database (https://busco.ezlab.org/). Details about the assemblies, isolates, and accession numbers are available in [57]. Nanopore sequencing reads are deposited in the European Nucleotide Archive (ENA) under the accession numbers ERR2612751 (BR32), ERR2612749 (FR13), ERR2612750 (US71), and ERR2612752 (CD156).

## Mini-chromosome isolation, library preparation and sequencing

Mini-chromosomes of *M. oryzae* were separated from core-chromosomes by contour-clamped homogenous electric field (CHEF) gel electrophoresis. Therefore, plugs containing the digested protoplasts (see above) were placed in a 1% megabase agarose gel. We used 0.5% TAE buffer for subsequent DNA extraction or 0.5% TBE buffer for visualization (Fig 1). We separated mini-chromosomes using a Biorad CHEF DRII system with the following settings: Run time: 96 h; Voltage: 1.8–2 V/cm; Initial switch interval: 120 s; End switch interval: 3600 s. The gel was then dyed with ethidium bromide solution (1 μg/ml) for visualization on a UV-transilluminator and individual mini-chromosome bands were excised.

Mini-chromosomal DNA was eluted from the gel plugs by electroelution using a D-Tube dialyzer midi, MWCO 3.5 kD (Merck). Therefore, individual plugs were placed in the dialysis tube and covered with 0.5% TAE buffer. The mini-chromosomal DNA was electroeluted for 3h at 90 V, resuspended by slowly pipetting up and down and concentrated in a vacuum centrifugal evaporator to a concentration of 50–150 ng/μl. Concentration of the extracted DNA was measured spectrophotometrically and fluorometrically by Nanodrop and Qubit, respectively, prior to library preparation.

Libraries for mini-chromosome sequencing were generated following the general guidelines of the Nextera Flex library preparation kit (Illumina). We modified the protocol as follows. For tagmentation, we used a total volume of 5 μl consisting of 2.5 μl Tn5 transposase, 0.5 μl of 10x reaction buffer, and 2 μl mini-chromosomal DNA set to a concentration of 0.5 ng/μl. The tagmentation mix was incubated in a thermocycler at 55°C for 7 min. The tagmentation product was then used in a PCR reaction containing 2.5 μl custom barcoding primers (10 μM) (S6 Table), 25 μl 2x NEBNext High-fidelity PCR mix (New England Biolabs), and 15 μl $H_2O$. PCR conditions: 72°C for 5 min, 98°C for 30 sec, 5 cycles of 98°C for 10 sec followed by 63°C for 30 sec and 72°C for 1 min. The numbers of amplification cycles needed for library preparation of each sample was then determined by quantitative PCR as follows: 25 μl SYBR Green Jumpstart Taq ReadyMic (Sigma-Aldrich), 5μl PCR product from previous reaction, 2.5 μl of each barcoding primer, 15 μl $H_2O$. PCR conditions as described above, without the initial 72°C step. Same PCR conditions were then applied to amplify the tagmented DNA. Libraries were then cleaned up using AMPure PB Bead purification kit (Pacific Biosciences). Concentration and quality of the libraries was confirmed by Nanodrop, Qubit, and BioAnalyzer 2100 using the high sensitivity DNA Kit (Agilent Technologies). Sequencing of mini-chromosomal DNA libraries was carried out on a NextSeq 500 system (Illumina) using the NextSeq 500/550 Mid output Kit (Illumina). Mini-chromosome derived reads were deposited at the European nucleotide archive under the accession numbers ERR3771227-ERR3771238.

## Identification of mini-chromosomes in whole genome assemblies

The quality of reads obtained from mini-chromosome isolation sequencing was confirmed with fastQC (https://www.bioinformatics.babraham.ac.uk/projects/fastqc/) and low quality sequences as well as adaptor sequences were removed using trimmomatic [81]. Mini-chromosome reads of each strain were then mapped against the whole genome assembly of the same strain using the BWA-mem (burrows-wheeler aligner) algorithm with default parameters. Reads were then filtered to keep only uniquely mapping reads using the samtools package [82] to prevent ambiguous mapping of reads originating from repeat rich regions. MCIS read coverage was calculated per 1 kb sliding window (window size: 1000 bp; slide: 500 bp) using the bedtools package (https://bedtools.readthedocs.io/en/latest/) and plotted using the R package circlize [83]. Mini-chromosome coverage and repeat content used to generate circus plots are shown in S7 Table.

Read mapping against the reference assembly version 8 of strain 70–15 (ensemble release version 45) and visualization was carried out as described above. The number of reads mapping to the reference genome was calculated for total reads and uniquely mapping reads. Uniquely mapping reads were used for visualization to exclude ambiguous mapping sites. Total amount of mapping reads is reported as percentage of total reads per sample. The supercontig number in assembly version 8 of isolate 70–15 corresponds to the chromosome number and supercontig_8.8 contains unmapped sequence.

## Whole genome and mini-chromosome alignments

Global alignments were generated using the nucmer algorithm of the MUMMER3 package [77]. The alignments were further filtered using the delta-filter utility (MUMMER3) with parameters -l 10000 and -i 80 (length >10 kb; percent identity >80%) to retrieve continuous alignments. Alignment coordinates were extracted with the show-coords utility and the output was modified to generate coordinate files in "bed" format for plotting in R. Plots were generated in R using the packages circlize [83], for circular representations of mini- and core-chromosome alignments, and karyoploteR [84], for linear representation of syntenic regions.

To calculate the breadth of coverage of whole genome alignments across several host specific lineages, all assemblies were aligned to the mini-chromosome contigs and selected core-chromosome contigs of the three isolates using nucmer. We then used the raw alignments and filtered alignments (length >5 kb) to generate coordinate files in "bed" format. Overlapping alignments were merged using bedtools v2.29.2 to generate non-redundant alignments for each mini-chromosome which were used to calculate the breadth of coverage relative to the total size of each mini-chromosome.

To calculate the breadth of coverage in the rice infecting lineage, each core chromosome or set of mini-chromosomes belonging to the same isolate was indexed using "bwa index". We retrieved all trimmed Illumina reads from Latorre *et al*., 2020 [44] and mapped them to the different indexed contigs using *bwa mem*. PCR duplicates were marked using *Picard Tools*. Finally, we used the samtools depth function to calculate the breadths of coverage for at least 1X, for every contig without filtering for Mapping Quality (MQ0). The overall pipeline description is available at https://gitlab.com/smlatorreo/magnaporthe_minichromosomes.

## Analysis of gene and repeat content

Gene models for all strains were retrieved from previous assemblies published via the *Magnaporthe* GEMO database ([48]; http://genome.jouy.inra.fr/gemo/). We then identified genes using a similarity-based approach. We used blastn [85] with all strain specific gene models to identify genes in our assemblies using a threshold of >90 query coverage and 99% identity to

account for differences in base calling quality and sequencing errors between two assemblies of the same strain. The >90 coverage threshold was chosen to account for errors in mononucleotide repeats that can occur from nanopore sequencing and the 99% identity threshold was empirically determined by whole genome alignments of both assemblies of the same strain which resulted in an average sequence identity between 99% and 99.5%. Resulting genes were assigned to mini- and core-chromosome encoded. Gene density was calculated as number of features per 10 kb window across each genomic compartment (core- and mini-chromosomes).

To analyze gene presence/absence of mini-chromosome encoded genes across host-specific lineages of *M. oryzae*, we extracted all gene models that are exclusively present on the mini-chromosomes and aligned them to 107 publicly available genome assemblies representing 10 genetic lineages of *M. oryzae* and the outgroups PM1 (*M. pennisetigena*) and six *M. grisea* isolates using minimap2. We then extracted all best hits with a query coverage of >90% to retrieve conserved genes and plotted their similarity to the query [%] and used python (3.8.3) to plot coverage values as a heatmap, using the pandas library (1.0.4) for data handling, and matplotlib (3.2.1) and seaborn (0.11.0) libraries for plotting. The seaborn function 'heatmap' was used for basic heatmap generation, and 'clustermap' was used for the clustered x-axis heatmap.

Repetitive sequences were annotated with RepeatMasker (http://www.repeatmasker.org/) using a merged library of repeats consisting of the RepBase repeat library for fungi (https://www.girinst.org/repbase/) and *Magnaporthe* repeats identified by Chiapello et al. [48]. Total repeat content for the whole genome as well as for mini- and core-chromosome scaffolds was extracted from RepeatMasker.

For bootstrapping of gene and repeat content, we extracted the coordinates of features from the gene model blastn and RepeatMasker output. We then transformed the data into bed format for analysis in R. For each mini-chromosome scaffold, we sampled 10,000 random regions from the core genome with equal size to the respective mini-chromosome scaffold using the randomizeRegions function of the regionR package. We then determined the numbers of genes and the basepairs occupied by repeats for each core-chromosome fragment and the mini-chromosome scaffolds using the countOverlaps function of the GenomicRanges package. Density plots of the frequency of features per core-chromosome fragment were generated with the R package ggplot.

## Prediction of secreted proteins

To predict secreted proteins, we retrieved the protein annotation of all strains from the *Magnaporthe* GEMO database ([48]; http://genome.jouy.inra.fr/gemo/). We then predicted proteins containing a signal peptide using the program SignalP v2.1 [86]. We then used TargetP-2.0 [87] to identify mitochondrial proteins and the hidden-markov model TMHMM [88] to identify proteins containing transmembrane domains. After removing mitochondrial and transmembrane proteins, we matched secreted proteins to the blastn output used to transfer gene models (described above) to determine the number and genomic location of secreted proteins in the nanopore/canu assemblies.

## Pfam domain annotation

Pfam domains were predicted by a hidden-markov model scan using the Pfam protein families database ([89]; https://pfam.xfam.org/). Protein sequences were retrieved from the *Magnaporthe* GEMO database as described above. We used a total of 772 non-redundant, mini-chromosome encoded proteins for the analysis. These contain all mini-chromosome encoded proteins after removal of duplicated, identical sequences.

## Predicting orthologous groups by TRIBE-MCL

Orthologous families were predicted using the software TRIBE-MCL [90]; http://micans.org/
mcl/). We combined the proteomes of all strains and identified homologous proteins using
blastp with a query coverage threshold of 90% and e-value of 10e^10. We then transform the
blastp output into an MCL readable format using the mcxdeblast command of TRIBE-MCL
and identify tribes by mcl and extracted tribes and protein identifiers using custom perl scripts.
Of 12951 tribes predicted from the combined proteome retrieved from the GEMO Magna-
porthe database, the members of 12900 matched the new assemblies with high coverage
(>90%) and identity (>99%). Resulting tribes were then categorized into three groups: i) tribes
that are conserved in all isolates, ii) tribes that are missing in one of the isolates, and iii) tribes
only present in one isolate. Then, we assigned the genomic location of each of the tribe mem-
bers as core-chromosome or mini-chromosome encoded, based on our mini-chromosome
identification.

## Supporting information

**S1 Fig. MCIS-read mapping and repeat content of *M. oryzae* strain FR13.** Circos plot of
mini-chromosome isolation sequencing (MCIS) coverage and repeat content across the FR13
genome. Outer ring (rainbow colors): Scaffolds and scaffold sizes. Outer track (Red/Black):
MCIS coverage per sliding window. Window size = 1000 bp; Slide distance: 500 bp. Y-axes:
average coverage per 1 kb window; axis limits set to min/max coverage. Inner track (Blue/
Black): Repeat content per sliding window. Window size = 10 kbp; Slide distance: 5 kbp. Y-
axes: repeat content in bp per 10 kb window; axis limits set to zero to maximum.
(TIF)

**S2 Fig. MCIS-read mapping and repeat content of *M. oryzae* strain US71.** Circos plot of
mini-chromosome isolation sequencing (MCIS) coverage and repeat content across the US71
genome. Outer ring (rainbow colors): Scaffolds and scaffold sizes. Outer track (Red/Black):
MCIS coverage per sliding window. Window size = 1000 bp; Slide distance: 500 bp. Y-axes:
average coverage per 1 kb window; axis limits set to min/max coverage. Inner track (Blue/
Black): Repeat content per sliding window. Window size = 10 kbp; Slide distance: 5 kbp. Y-
axes: repeat content in bp per 10 kb window; axis limits set to zero to maximum.
(TIF)

**S3 Fig. MCIS-read mapping and repeat content of scaffolds < 2 Mb in *M. oryzae* strain
US71. A)** Circos plot of mini-chromosome isolation sequencing (MCIS) coverage and repeat
content across US71 scaffolds < 2Mb. **B)** Circos plot of mini-chromosome isolation sequenc-
ing (MCIS) coverage and repeat content across US71 scaffolds < 200 kb. Outer ring (rainbow
colors): Scaffolds and scaffold sizes. Outer track (Red/Black): MCIS coverage per sliding win-
dow. Window size = 1000 bp; Slide distance: 500 bp. Y-axes: average coverage per 1 kb win-
dow; axis limits set to min/max coverage. Inner track (Blue/Black): Repeat content per sliding
window. Window size = 10 kbp; Slide distance: 5 kbp. Y-axes: repeat content in bp per 10 kb
window; axis limits set to zero to maximum.
(TIF)

**S4 Fig. MCIS-read mapping and repeat content of *M. oryzae* strain CD156. A)** Circos plot
of mini-chromosome isolation sequencing (MCIS) coverage and repeat content across the
CD156 genome. **B)** Circos plot of mini-chromosome isolation sequencing (MCIS) coverage
and repeat content across US71 scaffolds < 2Mb. Outer ring (rainbow colors): Scaffolds and
scaffold sizes. Outer track (Red/Black): MCIS coverage per sliding window. Window

size = 1000 bp; Slide distance: 500 bp. Y-axes: average coverage per 1 kb window; axis limits set to min/max coverage. Inner track (Blue/Black): Repeat content per sliding window. Window size = 10 kbp; Slide distance: 5 kbp. Y-axes: repeat content in bp per 10 kb window; axis limits set to zero to maximum.
(TIF)

**S5 Fig. Nanopore sequencing read coverage of proposed mini-chromosome scaffolds in US71.** The upper panel of each plot shows the nanopore read coverage per 1 kb sliding window across proposed mini-chromosome contigs. Lower panels show mini-chromosome sequencing coverage per 1 kb sliding window. The red line indicates the average coverage across the whole genome. The green line shows the average coverage per scaffold and the blue line shows the average coverage in regions of reduced coverage in scaffolds 14 and 21. The regions are indicated by the blue bars. Axes limits were manually set to best represent the data range in each plot.
(TIF)

**S6 Fig. The CD156 mini-chromosome contains duplicated sequences matching the subtelomeric region of core-chromosome scaffold 2. A)** Nanopore read coverage across scaffold 2 after mapping to the whole genome sequence (upper panel) and mini-chromosome read coverage (lower panel) in CD156. **B)** Nanopore read coverage across scaffold 2 after mapping to the isolated scaffold 2 (upper panel) and mini-chromosome read coverage (lower panel) in CD156. **C)** Alignment of CD156 mini-chromosome scaffolds to scaffold 2 (upper panel) and repeat content of scaffold 2 (lower panel). Axes limits were manually set to best represent the data in each plot.
(TIF)

**S7 Fig. MCIS reads of FR13, US71, and US71 mapped against the reference genome of strain 70–15. A)** Circos plot of MCIS reads uniquely mapped against the 70–15 genome. Outer ring: 70–15 chromosomes and chromosome sizes. Tracks: 1. FR13 MCIS read depth, 2. US71 MCIS depth, 3. CD156 MCIS read depth. **B)** Relative amount of MCIS total reads that mapped to the genome of strain 70–15. Mapped reads shown in green, unmapped reads in red.
(TIF)

**S8 Fig. Conservation of mini-chromosomal sequences within the rice-infecting genetic lineage of M. oryzae.** Breadth of coverage of each mini-chromosome scaffold and selected core-chromosome scaffolds after mapping of raw read data of 131 rice-infecting isolates. Isolate IDs are given on top of the columns. Scaffold IDs are given at the bottom. Left: Schematic representation of the rice-lineage phylogeny, adapted from Latorre et al., 2020. Sublineages are indicated by colors. The bottom row of the heatmap contains FR13 mapping data.
(TIF)

**S9 Fig. Conservation of FR13 mini-chromosome encoded genes.** The heatmap show sequence similarity and presence/absence of genes encoded on the FR13 mini-chromosome across 10 genetic lineages of M. oryzae. Mini-chromosome encoded genes are hierarchical clustered on the x-axis. Isolate IDs are shown on the y-axis. Host-specific lineages are indicated on the left.
(TIF)

**S10 Fig. Conservation of US71 mini-chromosome encoded genes.** The heatmap show sequence similarity and presence/absence of genes encoded on the US71 mini-chromosome across 10 genetic lineages of M. oryzae. Mini-chromosome encoded genes are hierarchical

clustered on the x-axis. Isolate IDs are shown on the y-axis. Host-specific lineages are indicated on the left.
(TIF)

**S11 Fig. Conservation of CD156 mini-chromosome encoded genes.** The heatmap show sequence similarity and presence/absence of genes encoded on the CD156 mini-chromosome across 10 genetic lineages of M. oryzae. Mini-chromosome encoded genes are hierarchical clustered on the x-axis. Isolate IDs are shown on the y-axis. Host-specific lineages are indicated on the left.
(TIF)

**S12 Fig. Known effector genes and predicted MAX-effectors are encoded on mini-chromosomes and can be duplicated between mini- and core-chromosomes. A)** Position of effector genes in the FR13 genome. **B)** Position of effector genes in the US71 genome. **C)** Position of effector genes in the CD156 genome. Characterized effector genes are shown in red and predicted MAX-effectors are shown in black throughout A-C. Duplications are shown as lines in the center. Mini-chromosome scaffolds are shown in red, core-chromosomes in grey. **D)** Copy numbers of known effector genes in the genomes of isolates FR13, US71, and CD56. Absence shown in grey and presence shown in blue. Duplicated and triplicated genes are shown in yellow and red, respectively. Numbers in cells show the percentage identity of individual copies.
(TIF)

**S13 Fig. Genome structure and telomeres in FR13.** Alignments >10 kb between FR13 and the reference genome 70–15 are shown as genetic links after reordering the FR13 scaffolds according to the genome structure of 70–15 using MAUVE. Colors indicate the matching chromosomes and scaffolds between 70–15 and FR13. Colored genomic links represent alignments in forward direction. Black genomic links represent inverted alignments. Dashed lines show completely assembled chromosomes in FR13. Telomeric repeats are indicated by arrowheads. Mini-chromosome scaffolds are shown in blue. The proposed mini-chromosome structure is shown in the top right corner.
(TIF)

**S14 Fig. Genome structure and telomeres in US71.** Alignments >10 kb between US71 and the reference genome 70–15 are shown as genetic links after reordering the US71 scaffolds according to the genome structure of 70–15 using MAUVE. Colors indicate the matching chromosomes and scaffolds between 70–15 and US71. Colored genomic links represent alignments in forward direction. Black genomic links represent inverted alignments. Telomeric repeats are indicated by arrowheads. Mini-chromosome scaffolds are shown in blue. The proposed mini-chromosome structure is shown in the top right corner. Note that, based on the current assembly, we cannot resolve the mini-chromosome of US71.
(TIF)

**S15 Fig. Genome structure and telomeres in CD156. A)** Alignments >10 kb between CD156 and the reference genome 70–15 are shown as genetic links after reordering the CD156 scaffolds according to the genome structure of 70–15 using MAUVE. Colors indicate the matching chromosomes and scaffolds between 70–15 and CD156. Colored genomic links represent alignments in forward direction. Black genomic links represent inverted alignments. Telomeric repeats are indicated by arrowheads. Mini-chromosome scaffolds are shown in blue. **B)** Alignments >10 kb between CD156 and the reference genome MZ5-6-1 are shown as genetic links after reordering the CD156 scaffolds according to the genome structure of MZ5-6-1 using MAUVE. Colors indicate the matching chromosomes and scaffolds between MZ5-6-1

and CD156. Colored genomic links represent alignments in forward direction. Black genomic links represent inverted alignments. Telomeric repeats are indicated by arrowheads. Mini-chromosome scaffolds are shown in blue. The proposed mini-chromosome structure based on the alignment to MZ5-6-1is shown in the top right corner.
(TIF)

**S16 Fig. Nanopore coverage supporting the observed rearrangement between 70–15 chromosome 2 and FR13 scaffold 1.** Nanopore coverage per 1 kb sliding window is shown in green bars. Alignments identified in Fig 6 are shown as red and blue rectangles.
(TIF)

**S17 Fig. The sequence content of the FR13 mini-chromosome is partially conserved in core-chromosomes in isolates from various host-specific lineages.** Raw alignment data of selected high quality whole genome assemblies against mini-chromosome scaffolds of FR13, US71 and CD156. X-axis: mini-chromosome scaffolds. Y-axis: whole genome assemblies of the isolates 70–15 (oryza lineage), MZ5-1-6 (eleusine lineage) and BR32 (triticum lineage). Dots show alignments. Lines indicate continuous alignments. Alignment color shows sequence similarity [%] between the query and the reference. Color scale = 80–100%.
(TIF)

**S1 Table. Summary of gene and repeat content.**
(XLSX)

**S2 Table. Conservation and genomic location of orthologous groups.**
(XLSX)

**S3 Table. Summary of Pfam domain analysis.**
(XLSX)

**S4 Table. Summary of secreted protein prediction.**
(XLSX)

**S5 Table. Alignments and read support for merging of contigs.**
(XLSX)

**S6 Table. Barcoding primers used for library preparation.**
(XLSX)

**S7 Table. Coverage of mini-chromosome derived reads and repeat content.**
(XLSX)

## Acknowledgments

We thank Elisabeth Fournier for providing the isolates used in this study, Florian Charriat and Pierre Gladieux for providing the repeat library used to annotate repetitive regions, Pingtao Ding for helpful advice and support during library preparation and mini-chromosome sequencing, Dan MacLean and Clémence Marchal for bioinformatics support, and Nick Talbot for critically reading the manuscript.

## Author Contributions

**Conceptualization:** Thorsten Langner, Sophien Kamoun.

**Data curation:** Thorsten Langner, Adeline Harant, Luis B. Gomez-Luciano, Ram K. Shrestha, Angus Malmgren, Sergio M. Latorre, Hernán A. Burbano, Joe Win.

**Formal analysis:** Thorsten Langner, Luis B. Gomez-Luciano, Ram K. Shrestha, Angus Malmgren, Sergio M. Latorre, Joe Win.

**Funding acquisition:** Sophien Kamoun.

**Investigation:** Thorsten Langner, Adeline Harant, Joe Win, Sophien Kamoun.

**Methodology:** Thorsten Langner, Sophien Kamoun.

**Project administration:** Thorsten Langner, Sophien Kamoun.

**Resources:** Thorsten Langner, Adeline Harant, Joe Win, Sophien Kamoun.

**Supervision:** Hernán A. Burbano, Sophien Kamoun.

**Visualization:** Thorsten Langner, Angus Malmgren, Sergio M. Latorre.

**Writing – original draft:** Thorsten Langner, Sophien Kamoun.

**Writing – review & editing:** Thorsten Langner, Luis B. Gomez-Luciano, Ram K. Shrestha, Angus Malmgren, Sergio M. Latorre, Hernán A. Burbano, Joe Win, Sophien Kamoun.

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
