## [Decision Letter · Decision Letter 0]

17 Mar 2020

Dear Dr. Kamoun,

Thank you very much for submitting your manuscript "Genomic rearrangements generate hypervariable mini-chromosomes in host-specific lineages of the blast fungus" (PGENETICS-D-20-00101) for review by PLOS Genetics. As with all papers peer reviewed by the journal, your manuscript was reviewed by members of the editorial board and by several independent peer reviewers. Based on the reports, we regret to inform you that we will not be pursuing this manuscript for publication at PLOS Genetics. The reviewers liked the topic, but felt that the manuscript in its current form lacked the depth and novelty that we seek at PLoS Genetics. 

The reviews are attached below this email, and we hope you will find them helpful if you decide to revise the manuscript for submission elsewhere. We are sorry that we cannot be more positive on this occasion.

While we cannot consider your manuscript further for publication in PLOS Genetics, we would like to offer you the option to transfer your submission, with reviews, to PLOS ONE https://www.editorialmanager.com/PONE/
.

If you DO wish to transfer your submission, please click this link:

<DeepLinkData><DeepLinkTypeID>27</DeepLinkTypeID><peopleID>244657</peopleID><userSecurityID>03bc6197-4ae8-4014-bc19-fbd5e8dfd848</userSecurityID><documentID>33949</documentID><revision>0</revision><manuscriptNumber>PGENETICS-D-20-00101</manuscriptNumber><docSecurityID>8f02e4ee-c167-4746-a35a-0b4f6cd00e39</docSecurityID></DeepLinkData>

If you do NOT wish to transfer your submission, please click this link to decline:

<DeepLinkData><DeepLinkTypeID>28</DeepLinkTypeID><peopleID>244657</peopleID><userSecurityID>03bc6197-4ae8-4014-bc19-fbd5e8dfd848</userSecurityID><documentID>33949</documentID><revision>0</revision><manuscriptNumber>PGENETICS-D-20-00101</manuscriptNumber><docSecurityID>8f02e4ee-c167-4746-a35a-0b4f6cd00e39</docSecurityID></DeepLinkData>

Please note, all PLOS journals are editorially independent and vary in submission requirements.

Should you choose to transfer, your manuscript files, along with the reviewers' comments and their identities will be transferred automatically, and you will receive a confirmation email within 24 hours. Once transferred, your submission will be returned to you so you can check over your record before completing the submission. You may be asked to provide additional information, such as a response to the reviewers' comments. If you have any questions, please contact the editorial office of PLOS ONE https://www.editorialmanager.com/PONE/
.

We are sorry that the news is not more positive on this occasion and hope that you will consider PLOS Genetics for other submissions in the future. Thank you for your support of PLOS and of open-access publishing.

Sincerely,

Xiaorong Lin, Ph.D.

Associate Editor

PLOS Genetics

Hua Tang

Section Editor: Natural Variation

PLOS Genetics

Reviewer's Responses to Questions

**Comments to the Authors: **

Reviewer #1: The manuscript “Genomic rearrangements generate hypervariable mini-chromosomes in host-specific lineages of the blast fungus” sequenced four field Magnaporthe oryzae isolates from four host species with the Nanopore long-read technology, producing four near finished genome assemblies. With the results from CHEF and MCIS, contigs derived from mini-chromosomes were identified. Characterization of sequences of mini-chromosomes found that they have lower levels of gene content and higher levels of repeats as compared to core-chromosomes, and mini-chromosomes seemed to have lineage-specific sequences. In addition, mini-chromosomes contain virulence-related loci, which are likely resulted from genomic rearrangements from core-chromosomes. The results from this study largely agree with the findings in a recent mini-chromosome paper, Peng et al., 2019 (reference 56). Most conclusions in the manuscript were well supported by data. The major concern for this manuscript is related to the speculation that mini-chromosomes emerged independently in different lineages, which was based on high levels of dissimilarity among mini-chromosomes from M. oryzae isolates from three host species. The high-level diversity of mini-chromosomes could be due to low levels of selective pressure, which allows high frequencies of base mutations and genomic rearrangements. It is not clear why independent emergence is necessary for the high-level diversity. Some detailed comments are listed below:

L195-196: 17-55 are scaffold numbers rather than contig numbers 

S1 fig: almost no blue color can be observed. Suggest adding notes on the figure to label each track or do other changes to make it clear.

About the hypothesis that US71 contains two near-identical mini-chromosomes, is it possible that some mini-contigs (identified as sequences from the mini-chromosome) are chimeric contigs from both core- and mini-chromosome? Enlargemet of S2 and S3 figures might provide the clue or reject this possibility. 

P251-252: Again, this need to consider the possible assembly error.

P274-275: “~900kb aligned on chr2 of 70-15” should be at the end of chr2 based on S5 fig. This is an interesting result, which should be mentioned here although it was described in the next paragraph.

P287-293: The description is not clear. What does “largely overlapping regions” mean?

P304-307: The speculation is not well supported. Data indicated that mini-chromosomes are hypervariable. Not population data, particularly data from diverse isolated per lineages, here support that mini-chromosomes emerged independently in host-adapted lineages. 

P310-312: To map annotated genes to a new genome is not a satisfied approach for genome annotation. But I guess the authors just wanted to find homologs to an annotated gene database.

P339-343: I don’t understand the reason for bootstrapping analysis. Data using all gene/repeats contents from all core and mini-chromosomal data are convincing. I don’t think bootstrapping is necessary.

P396-397: It is interesting that the number of secreted proteins varies dramatically among three strains. The numbers of genes on mini-chromosomes cannot completely explain this variation. I am just curious what implication this could provide.

L412-413: Assemblies of mini-chromosomes in this study cannot exactly know where are ends of mini-chromosomes, right? This statement probably needs to be revised.

Which version of 70-15 reference genome was used? Is Supercontigs8.2 chr2?

Reviewer #2: The manuscript “Genomic rearrangements generate hypervariable mini-chromosomes in host-specific lineages of the blast fungus” by Thorsten Langner and colleagues described mini-chromosomes in four lineages of the blast fungus. In agreement with many reports from other pathosystems, such as Fusarium, that these mini-chromosomes differ from core chromosomes with all distinct characteristics: high repeat, low gene content, and lack house-keeping genes.

The discovery is interesting, but not quite novel. Even though different terms were used, mini-chromosomes had been a focus for many other pathosystems, such as Zymoseptoria and Fusarium just to name a couple. Also, a paper published in PLOS Genet 2019 “Effector Gene Reshuffling Involves Dispensable Mini-chromosomes in the Wheat Blast Fungus” by Dr. Valent and colleagues already extensively documented such a phenomenon in a blast fungal genome . Unfortunately, this work was not even cited in this manuscript and the mini-chr reported in that paper was not included in the comparison. 

The author concluded that mini-chromosomes were emerged from structural rearrangements of core-chromosomes. I feel this conclusion is not solidly supported. Here are two reasons:

1) Comparison between mini-chr with core-chr revealed some similarity, but more than 50% are not accountable. 

a total of 42,974 genes between all three isolates, which can be formed into 12,900 orthogroups.

 9,003 were conserved across all three 

 3,897 were absent in at least one strain 

 1,827 that were conserved between two 

 2,070 were unique to single strains. 

 50% have no functional annotation 

2) current data can’t differentiate two hypotheses of a) two mini-chromosome of the same size b) segmental duplication and translocation based on the observation of increased coverage of CD156. If second hypothesis is correct (based on our experience, it is very possible), then the match could be the reintroduction of mini-chr content to the core, in the contrast of the conclusion presented by the authors. 

This definitely needs to investigate further.

Reviewer #3: The authors assess the mini-chromosomes in several lineages of M. oryzae. Large fragment genomic sequences were obtained by nanopore sequencing, and mini-chromosome specific reads obtained from CHEF gel separated chromosomes. Three of the four host specific lineages of M. oryzae had mini chromosomes, which were found to have several characteristics significantly different than the “core” chromosomes including a reduced gene density and an increase in repetitive sequences. These mini-chromosomes appear to be enriched in genes encoding putative virulence factors. Synteny analysis indicated regions of synteny around the ACE1 locus, a known virulence factor; however the lack of conservation of these mini-chromosomes led to speculation that they arose independently. 

While the improved assembly and reanalysis of the four M. oryzae genomes is a nice addition, overall the manuscript adds little new information to what was previously known about supernumerary chromosomes and lacks experimental support. Are any of these mini-chromosomes able to be lost? Do they influence the host-specificity of each of the lineages? 

Other concerns:

1. Do other field isolates in the same host-specific lineages have the same (or similar) mini-chromosomes? Sequencing the CHEF-extracted mini-chromosomes would add support for many of these conclusions.

2. Why were only 772 non-redundant, mini-chromosome encoded proteins included in the Pfam domain search (page 31, line 375)? 

3. With the emphasis on secreted proteins and their possibility to serve as effectors, 50, 8, and 2 secreted proteins were found on the mini-chromosomes in the three host specific lineages (page 32, lines 369-398). Are any of these actually expressed during infection? Similarly, there were 7 MAX-effectors on the mini-chromosomes (page 34, lines 407-409) – are any of these expressed?

4. It would be nice to include a description of what was in the 392 kb region separating the regions of synteny around the ACE1 locus within FR 13. Is this mostly repetitive sequence? 

5. Additionally, what is encoded within the 761 kb and 164 kb mini synteny regions around the ACE1 locus? Is there anything specifically that could be contributing to host-specificity?

**Have all data underlying the figures and results presented in the manuscript been provided?**

Reviewer #1: Yes

Reviewer #2: Yes

Reviewer #3: Yes

PLOS authors have the option to publish the peer review history of their article (what does this mean?). If published, this will include your full peer review and any attached files.

Reviewer #1: No

Reviewer #2: No

Reviewer #3: No

---

## [Decision Letter · Decision Letter 1]

1 Jul 2020

Dear Dr Kamoun,

Thank you very much for submitting your Research Article entitled 'Genomic rearrangements generate hypervariable mini-chromosomes in host-specific lineages of the blast fungus' to PLOS Genetics. Your manuscript was fully evaluated at the editorial level and by independent peer reviewers. Their comments are attached below. Based on the reviews (particularly comments from reviewer 3), we would be willing to consider a much revised version, with some experimental evidence to support the role of the minichromosomes. We cannot, of course, promise publication at that time.

If you decide to revise the manuscript for further consideration at PLOS Genetics, please aim to resubmit within the next 60 days, unless it will take extra time to address the concerns of the reviewers, in which case we would appreciate an expected resubmission date by email to plosgenetics@plos.org.

[LINK]

We are sorry that we cannot be more positive about your manuscript at this stage. Please do not hesitate to contact us if you have any concerns or questions.

Yours sincerely,

Xiaorong Lin, Ph.D.

Associate Editor

PLOS Genetics

Hua Tang

Section Editor: Natural Variation

PLOS Genetics

Reviewer's Responses to Questions

**Comments to the Authors:**

Reviewer #1: The authors have well addressed most of my questions/comments. I still do not fully agree the following statement and appreciate the consideration about some levels of similarity between mini-chromosomes.

Here is the statement in the revision: “The difference in sequence similarity between core- and mini-chromosomes supports the hypothesis that mini-chromosomes represent isolate-specific genomic entities, possibly involved in host specialization and suggests that mini-chromosomes emerged independently in host-adapted lineages or individual isolates.”

Reviewer #2: The authors have address all my concerns. No

Reviewer #3: In this revised submission concerning mini-chromosomes in Magnaporthe oryzae, the authors indicate that their original submission was in respect to the PLOS “scooping” policy and have addressed some of the original concerns in this resubmitted manuscript. The manuscript has improved with additional analyses, and does provide complementary information in support of the Peng at al publication in regards to the “scooping” policy.

As mentioned in the previous review, the manuscript adds little new information regarding the general biology of fungal supernumerary chromosomes. In the absence of additional experiments, it is difficult to assess the contribution of these chromosomes to virulence, in particular host specificity.

**Have all data underlying the figures and results presented in the manuscript been provided?**

Reviewer #1: Yes

Reviewer #2: Yes

Reviewer #3: Yes

PLOS authors have the option to publish the peer review history of their article (what does this mean?). If published, this will include your full peer review and any attached files.

Reviewer #1: No

Reviewer #2: No

Reviewer #3: No

---

## [Decision Letter · Decision Letter 2]

26 Jan 2021

Dear Dr Kamoun,

Thank you for your effort in addressing reviewers' concerns. We are pleased to inform you that your manuscript entitled "Genomic rearrangements generate hypervariable mini-chromosomes in host-specific isolates of the blast fungus" has been editorially accepted for publication in PLOS Genetics. Congratulations!

Yours sincerely,

Xiaorong Lin, Ph.D.

Associate Editor

PLOS Genetics

Hua Tang

Section Editor: Natural Variation

PLOS Genetics

Comments from the reviewers (if applicable):

Reviewer's Responses to Questions

**Comments to the Authors:**

Reviewer #1: The revision has been improved, which better reflects genomic data collected than the previous version. I would suggest accepting the manuscript. "ACE1" in P535 needs to be corrected to be consistent.

Reviewer #2: I appreciate authors efforts in improving the manuscript. Well done.

Reviewer #3: In the revised manuscript the authors have included a significant amount of additional comparative analyses, in particular including several genomes from other lineages that are virulent on other plants. While experimental evidence to support the role of the mini-chromosomes was not included, overall the authors have addressed all my major concerns.

**Have all data underlying the figures and results presented in the manuscript been provided?**

Reviewer #1: Yes

Reviewer #2: Yes

Reviewer #3: Yes

PLOS authors have the option to publish the peer review history of their article (what does this mean?). If published, this will include your full peer review and any attached files.

Reviewer #1: No

Reviewer #2: No

Reviewer #3: No

**Data Deposition**

http://datadryad.org/submit?journalID=pgenetics&manu=PGENETICS-D-20-00101R2

**Press Queries**

---

## [Editor Report · Acceptance letter]

10 Feb 2021

PGENETICS-D-20-00101R2 

Genomic rearrangements generate hypervariable mini-chromosomes in host-specific isolates of the blast fungus 

Dear Dr Kamoun, 

We are pleased to inform you that your manuscript entitled "Genomic rearrangements generate hypervariable mini-chromosomes in host-specific isolates of the blast fungus" has been formally accepted for publication in PLOS Genetics! Your manuscript is now with our production department and you will be notified of the publication date in due course.

With kind regards,

Alice Ellingham

PLOS Genetics

On behalf of:
